



# Evaluation of a High-Resolution Regional Climate Simulation for Surface and Hub-height Wind Climatology over North America

Kyle Peco[1], Jiali Wang[1], Chunyong Jung[1], Gökhan Sever[1], Lindsay Sheridan[2], Jeremy Feinstein[1], Rao Kotamarthi[1], Caroline Draxl[3a], Ethan Young[3], Avi Purkayastha[3], and Andrew Kumler[3]

[1]Environmental Science Division, Argonne National Laboratory, Lemont, IL, 60439, United States
[2]Pacific Northwest National Laboratory, Richland, Washington, 99354, United States
[3]National Renewable Energy Laboratory, Golden, Colorado, 80401, United States
[a]Now at Electric Power Research Institute, Palo Alto, California, 94304, United States

*Correspondence to*: Kyle Peco (kpeco@anl.gov) and Jiali Wang (jialiwang@anl.gov)

**Abstract.** Assessing the availability of key wind resources requires augmenting observations to support the implementation of wind energy infrastructure. However, observations are limited, necessitating the development of high resolution, long-term gridded datasets. This study presents a robust, dynamically downscaled climatological dataset, offering 20 years of hourly wind data at a 4-km spatial resolution across North America, and evaluates its performance against observations, including

meteorological towers and Automated Surface Observing Stations (ASOS), as well as a coarse-resolution reanalysis data — European Centre for Medium-Range Weather Forecasts (ECMWF) reanalysis version 5 (ERA5). Results demonstrate that the downscaled high resolution wind data outperforms ERA5 in regions of complex terrain and coastal areas, with improved overlap coefficients for wind data distributions and reduced root mean square errors (RMSE) for hub-height and near-surface diurnal wind patterns. The downscaled simulation also reasonably captures the synoptic drivers of seasonal wind direction

patterns, indicated by high wind rose similarity indices. This study also provides an analysis of interannual variability, utilizing the dataset's full 20-year period, and model uncertainty, generated by varying model initial conditions and physics parameterizations across 1-year ensemble members, which are key considerations for wind resource assessment in wind farm development.

## 1 Introduction

Wind is a key factor in shaping a region's complex climate, influencing both environmental and economic sectors. Understanding local and regional wind variability is vital for assessing wind energy potential, which aids in the efficient implementation and operation of wind farms (Millstein et al., 2019; Couto & Estanquiero 2021). Additionally, evaluating wind speed and direction is essential for conducting accurate risk assessments for high winds, whether onshore or offshore (Li, 2023;

Grasu & Liu, 2023; Wu et al., 2022b). However, the spatiotemporal coverage of current wind measurements remains very



limited, particularly over complex terrains (e.g., western US), offshore, and at hub-heights, where wind energy resource assessments are crucial.

To bridge the gap between limited observational data and the need for accurate wind resource assessments, global and regional reanalysis datasets, such as Modern-Era Retrospective analysis for Research and Applications version 2 (MERRA-2), the North American Regional Reanalysis (NARR), and the European Centre for Medium-Range Weather Forecasts Reanalysis version 5 (ERA5), are commonly used (Hersbach et al., 2020; Gelaro et al., 2017; Mesinger et al., 2006). These reanalysis datasets provide valuable insights into wind patterns, variability, and long-term trends, and are also crucial for capturing climatological oscillations and large-scale circulations that influence wind characteristics (e.g., Sheridan et al., 2022a). While these datasets typically have higher horizontal resolution than global climate models (GCMs), they still lack the resolution necessary to explicitly resolve convection, which is essential for capturing convectively driven precipitation and wind (Murakami, 2014; Jones et al., 2021). Additionally, validating these reanalysis datasets is essential for determining their viability for wind resource assessments. (Sheridan et al., 2020, 2024; Lee et al., 2014). For example, Sheridan et al. (2022b) found that ERA5 generally underestimates wind speed diurnal cycles based on 62 sites at a variety of heights above ground across the continental United States (CONUS). This underestimation is most prominent in late afternoon, caused primarily by the underestimation of convectively driven strong winds. Similarly, Chen at al. (2024) and Wilczak et al. (2024) found that ERA5 showed significant negative biases for wind speeds in areas of complex terrain, especially over the Rocky Mountains.

To achieve the necessary high resolution to capture finer scale wind patterns over large spatial areas and extended time periods, researchers employ a technique called dynamical downscaling. This technique involves using initial and boundary conditions from the global or regional reanalysis data to force simulations at finer resolutions using a regional climate model. Regional climate modeling at a convection-permitting (CP) resolution, with a horizontal grid spacing of less than approximately 4 km, has become a promising approach for delivering more reliable climate information at regional and local levels. By directly resolving deep convective processes rather than relying on parameterization, these models demonstrate significant enhancements (e.g., Prein et al., 2015 and the references therein). Due to recent breakthroughs in computational capacity and data management, several studies have been able to perform convection-permitting regional climate model (RCM) simulations. These simulations, especially those concentrating on the CONUS, (e.g., Draxl et al., 2015b; Gensini et al., 2023, Liu et al., 2017; Rasmussen et al., 2024) have shown substantial progress in depicting precipitation, wind, and high-impact weather from national to regional spatial scales. Among these, Draxl et al. (2015a, b) presented the largest, freely available wind dataset at the time of its creation, serving the Wind Integration National Dataset (WIND) Toolkit for wind resource assessment and grid integration studies. The data provides time series of meteorological variables every 5 min and 2km across the CONUS in the 7 years from 2007 to 2013.

This study builds upon previous efforts by presenting an additional high-resolution, long-term dataset, along with ensemble simulations for quantifying model uncertainty, for utilization in climatological wind assessments. The dataset was





generated by a regional climate model using the Weather Research and Forecasting (WRF) model. With 4 km, 20-year, hourly output, and a model domain spanning the majority of North America and surrounding oceans, this dataset provides a

spatiotemporal extension to existing climatological wind analyses. With large geographic coverage, this data product also offers insight into more remote, topographically complex regions, and high-impact weather phenomena, such as tropical and extratropical cyclones (TCs/ETCs) and atmospheric rivers, potentially highlighting viable areas for wind energy and providing the means for climate related risk assessments outside of CONUS. By leveraging a single large spatial domain, the model evolved as one system, developing its own natural variability without being constrained by the forcing data. This dataset has

been leveraged by the latest WIND Toolkit Long-term Ensemble Dataset (WTK-LED), as documented by Draxl et al. (2024), serving as the WTK-LED Climate dataset (Table ES-1 in Draxl et al. 2024). Ultimately, this high-resolution dataset aims to combine the climatological significance of an extensive temporal length with the wind-resource-utility advantages of a large spatial domain.

Our study validates the dynamically downscaled model wind speeds against various observational data at both the

near-surface and at turbine-heights at mostly inland and onshore locations, investigating model performance at different temporal scales (diurnal, seasonal, interannual variability). A complementary study evaluating the same dataset but focusing on CONUS coastal areas has been documented by Sheridan et al. (2024). Our validation is also performed on the forcing data —ERA5 reanalysis (Hersbach et al., 2020), aiming to understand the added value of the dynamically downscaled model to its coarser resolution forcing data. Additionally, this study seeks to augment insights on model uncertainty within wind

simulations that are brought about by varying model configurations.

This manuscript is organized in the following structure: the methodology, including model description, observational datasets used for validation, and analysis metrics used for evaluation are outlined in Section 2. The results of the model's performance at hub-heights and near surface are presented in Section 3.1 and 3.2, with an exploration of model bias in Section 3.3. Interannual variability and model uncertainty are quantified in Section 3.4 with the context of wind energy implications.

Lastly, a summary of our findings and avenues for future research are discussed in Section 4.

## 2 Methods and Datasets

### 2.1 Model Description

The wind validation performed in this study was based on a 20-year (2001-2020) climatological dataset produced by the WRF

model (Powers et al., 2017) version 4.2.1 with the Advanced Research WRF dynamic core (Skamarock & Klemp, 2008): the Argonne Downscaled Data Archive version 2 (ADDA-v2). With a domain of 2050 x 1750 grid points at a 4-km grid spacing (8200 km x 7000 km), the model featured over 3.5 million grid cells, horizontally spanning across the majority of North America and the Caribbean Islands (Fig. 1a in Akinsanola et al., 2024). The model was run with 50 unevenly spaced sigma





levels, 18 of which were within the lowest 1 km. The first six layers are below 104 meters above ground level — 8, 25, 41, 58,
75 and 104 meters. Initial and lateral boundary conditions were determined by ERA5; 5 variables were taken at each of ERA5's
37 available pressure levels alongside 26 single-level variables. The model was reinitialized for each year, ultimately producing
a series of 20, 14-month simulations covering the period from 2001-2020. These individual simulations were allowed a spin-
up period of two months (November and December), which were eventually discarded and not used for the data analysis. To
study the model's internal variability, we conducted 10 additional 1-year (ENSO neutral year - 2018) ensemble runs, all with
the same model setup, but different initial conditions (Wang et al., 2018). This was achieved by running each of the ten
ensemble members 12 hours apart, with the first being initialized on November 1, 2017, at 00 UTC and the last being initialized
on November 5, 2017, at 12 UTC. Thus, the slightly different initial conditions at each respective start time acted as the catalyst
to generate differences between the ensemble members.

Generally, WRF simulations employ multiple physics schemes to implicitly represent the sub-grid processes
occurring within the model domain. The choices specified for different parameterizations can impact how the model simulates
wind, especially PBL, surface layer schemes, and land surface processes. For the 20-year simulations the Yonsei University
(YSU) (Hong et al., 2006) PBL scheme is used, which has been studied in multiple model sensitivity experiments that explore
the effects of PBL schemes on wind simulations (Carvalho et al., 2012; Carvalho et al., 2014; Li et al., 2021; Wu et al., 2022a;
Hahmann et al., 2015; Draxl et al., 2014). Overall, the YSU PBL scheme performs relatively better in unstable conditions than
stable conditions and represents diurnal variability well (Hong et al., 2006; Draxl et al., 2014). The surface layer scheme was
the MM5 similarity scheme, which follows the Monin-Obukhov similarity theory (Monin & Obukhov, 1954) alongside the
Carlson-Boland similarity functions (Carlson and Boland, 1978). The Unified Noah land-surface model was used for the land
surface processes, which employs a 4-layer soil temperature and moisture scheme, as well as fractional snow cover and frozen
soil physics (Tewari et al., 2004). A full list of the model parameterizations can be found in Table 1.

To investigate the model's structure uncertainty arising from key physics parameterizations — namely the PBL and
land surface model (LSM) — an additional six ensemble members were generated for the same neutral year 2018. Each
ensemble member shared the same domain and spatial resolution but employed different PBL schemes (YSU and MYNN) and
LSMs (Noah and NoahMP). The MYNN PBL scheme is a level 2.5 closure scheme for turbulence and implicitly solves for
turbulence using parametric equations. It gives estimates of TKE and dissipation rates within the boundary layer of the
atmosphere (Nakanishi & Niino, 2009). Noah-MP is an improved version of the Noah LSM and provides better representations
of terrestrial biophysical and hydrological processes (Niu et al., 2011). Major physical mechanism enhancements include
improved treatment of soil moisture. No internal grid nudging, nor spectral nudging was employed for these simulations to
allow the model to develop its own spatiotemporal variability. Model output data for the most used meteorological variables,
such as air temperature, wind speed and direction, and precipitation, were saved at hourly intervals for the full domain from
2001-2020. Other variables less frequently used were saved at 3-hour intervals.



**Table 1:** WRF model setup and ensemble runs used in ADDA_v2 simulations

| Regional Climate Model | WRF v4.2.1 |
|---|---|
| **Initial and Boundary Conditions** | ERA5 at 0.25 deg, every 3 hours |
| **Horizontal Grid Spacing and Timesteps** | 4km; adaptive time stepping |
| **Number of Grid Cells** | 2050 (west-to-east) x 1750 (south-to-north) x 49 (top-to-bottom) |
| **Simulation Period** | January 1, 2001, to December 31, 2020 |
| **Microphysics Scheme** | Morrison double moment (Morrison et al., 2005) |
| **Land Surface Scheme** | Unified Noah (Tewari et al., 2004), Noah-MultiParameterization (NoahMP, Niu et al., 2011) with two options for dynamic vegetation and surface |
| **Planetary Boundary Layer Scheme** | Yonnsei University (Hong et al., 2006), Mellor-Yamada-Nakanishi-Niino (MYNN, Nakanishi & Niino, 2009) |
| **Short and Long-wave Radiation Scheme** | Rapid Radiative Transfer Model for GCMs (RRTMG; Iacono et al., 2008) |

## 2.2 Observational Datasets Used for Validation

The validation performed on ADDA-v2 used wind speed observational data taken within 100 meters above ground level. The first collection of observations focused on hub-height wind speeds and wind directions. These observations were taken from multiple meteorological towers hosted by the US Department of Energy National Laboratories (Argonne National Laboratory, Brookhaven National Laboratory, NREL, Oak Ridge National Laboratory, Pacific Northwest National Laboratory, Savannah River National Laboratory), and the National Oceanic and Atmospheric Administration (National Centers for Environmental

Information, National Data Buoy Center). In total, 26 meteorological towers were sampled and quality controlled for this analysis, with wind speed observations taken anywhere from 10m to 100m above ground level. Observations were quality controlled through the process of removing atypical or unphysical reported wind speeds (less than 0 m s$^{-1}$, greater than 50 m s$^{-1}$, or non-varying values over periods of time greater than 3h), based on Sheridan et al. (2024). Temporal coverage varied between 2-20 years, with an average of ~8.1 years. Observations covered a diverse range of geographies, including mountainous, coastal

(east and west coast of the CONUS), the Great Lakes, and plains regions; Alaska and Puerto Rico (Caribbean) were denoted as separate geographic regions. For 19 of these meteorological towers, the exact locations, anemometer heights, and temporal coverages of wind observations can be found in Table 2. The remaining 7 are proprietary data, in which exact locations could not be specified. While turbine-height wind speed and wind direction data is sparse, we have leveraged all the publicly available resources that we have access to and performed a thorough validation over diverse geospatial areas.


**Figure 1.** Locations of in-situ observations sampled from meteorological towers across CONUS and Alaska, along with an ASOS location over Puerto Rico. The zoomed in area, with stars representing each dataset, indicates the capability of ADDA-v2's higher resolution to more closely match the exact location of the in-situ data. The 2000+ sites over CONUS are not included here but can be seen in Fig. 6.

The second part of this evaluation explores an expansive collection of 10 m wind speed data sourced from a network of Automated Surface Observing Stations (ASOS). These stations monitor and report various meteorological variables and are operated by the United States National Weather Service, the Federal Aviation Administration, and the Department of Defense. The specific dataset used for this validation was collected from the Iowa Environmental Mesonet (IEM) and subsequently

quality controlled by the Data Archive and Portal (DAP) Platform. The dataset hosts over 2,000 sites across CONUS and





Alaska and covers a temporal period from 1 January 2000 – 31 December 2021, offering a spatiotemporally comprehensive means for performing a thorough validation of ADDA-v2's 10m wind. Additionally, wind speed data from four additional ASOS stations over Puerto Rico were downloaded from the Iowa Environmental Mesonet (IEM) to spatially expand the model validation and gain a more comprehensive understanding of model performance over areas of sparse data availability and complex terrain.

To demonstrate the potential added value of ADDA-v2 to its coarse resolution forcing data, we also include ERA5 reanalysis in all near-surface and hub-height evaluations. ERA5 outputs only two levels of wind (10m and 100m), so to evaluate winds at heights between these levels, an interpolation method was required. At each timestamp, the ADDA-v2 and ERA5 wind speeds were adjusted to the observational heights via the power law using the model wind speeds at surrounding output heights to the observation height. While this interpolation method may induce some bias in both ADDA-v2 and ERA5, the differences between these datasets are driven mostly by the difference in spatial resolution and the added value by ADDA-v2. This approach was selected based on the analysis of Duplyakin et al. (2021), who found that the power law minimized errors due to vertical adjustment of wind dataset output heights to observation heights.

## 2.3 Statistics for Validation

The wind speed validation in this study utilizes several statistical error metrics to evaluate how well ADDA-v2 performs against observations. In particular, root mean square error (RMSE), Pearson correlation coefficients ($r$), overlap coefficients (OVLs), and wind rose similarity indices (WRSIs) are used.

The RMSE gives a metric for the overall accuracy of the model, with lower RMSE's indicating improved model performance. RMSE is taken as the square root of the average of the squared differences between simulated wind speeds and the observed wind speeds at various timescales (seasonal, monthly, diurnal), given by Eq. (1). This metric is effective at highlighting instances of larger errors in the model and demonstrates the overall magnitude of model inaccuracy. Here, $n$ represents the number of wind speed observations (in time), $v_{\mathrm{mod}}$ represents the modeled wind speed, and $v_{\mathrm{obs}}$ denotes the observed wind speed. Relative RMSE (rRMSE) was also considered, Eq. (2), by dividing the RMSE by the average of the observed wind speed. This gives a general sense of the magnitude of error in relation to the magnitude of the wind speeds themselves.

$$\mathrm{RMSE} = \sqrt{\tfrac{1}{n}\sum_{i=0}^{n}(v_{\mathrm{mod},i} - v_{\mathrm{obs},i})^2} \tag{1}$$

$$\mathrm{rRMSE} = \frac{\mathrm{RMSE}}{\bar{v}_{\mathrm{obs}}} \tag{2}$$



**Table 2.** Information for the hub-height wind data sourced from meteorological towers across CONUS. The number listed for each location corresponds to the numbers in Fig. 1, identifying the geographic positions of the meteorological towers. Location coordinates for proprietary data were excluded.

| Geography | Location | Coordinates | Temporal Coverage | Anemometer Height |
|---|---|---|---|---|
| W. Coast | Megler, WA (1) | 46.27°N, -123.88°W | 2010-2018 | 53m |
| | Martinez, CA (3) | 38.04°N, -122.12°W | 2014-2020 | 100m |
| | Los Angeles Pier J, CA (4) | 33.73°N, -118.19°W | 2014-2020 | 31m |
| Mountain | Wasco, OR (2) | 45.50°N, -120.77°W | 2005-2018 | 30m |
| | NWTC, CO (5) | 39.91°N, -105.24°W | 2002-2020 | 50m |
| Plains | Site A, KS (6) | - | 2006-2008 | 49m |
| | SGP Observatory, OK (7) | 36.61°N, -97.49°W | 2012-2020 | 65m |
| | Site A, TX (8) | - | 2008-2013 | 50m |
| | Site B, TX (9) | - | 2009-2013 | 51m |
| | Site A, MN (10) | - | 2007-2011 | 80m |
| | Site A, AR (11) | - | 2011-2012 | 53m |
| | Argonne National Lab, IL (12) | 41.70°N, -87.99°W | 2007-2013 | 60m |
| | Site A, IN (13) | - | 2018-2019 | 90m |
| | Site A, OH (14) | - | 2017-2018 | 90m |
| Great Lakes | Dunkirk, NY (17) | 42.49°N, -79.35°W | 2001-2017 | 20m |
| E. Coast | Edith Hammock, AL (15) | 30.23°N, -88.02°W | 2008-2013 | 36m |
| | Fowey Rock, FL (16) | 25.59°N, -80.09°W | 2001-2020 | 44m |
| | Spiderweb, SC (18) | 33.41°N, -81.83°W | 2009-2012 | 34m |
| | East Point, FL (19) | 29.41°N, -84.86, °W | 2004-2020 | 35m |
| | Cape Henry, VA (20) | 36.93°N, -76.01°W | 2007-2020 | 28m |
| | Brookhaven, NY (21) | 40.87°N, -72.89°W | 2007-2013 | 50m |
| Alaska | Red Dog Dock, AK (22) | 67.58°N, -164.07°W | 2018-2020 | 13m |
| | Bligh Reef, AK (23) | 60.84°N, -146.88°W | 2013-2020 | 22m |
| | Juneau Dock, AK (24) | 58.29°N, -134.39°W | 2018-2020 | 18m |
| | Five Fingers, AK (25) | 57.27°N, -133.63°W | 2013-2020 | 22m |
| Puerto Rico | San Juan, PR (26) | 18.43°N, -66.01°W | 2001-2020 | 10m |



The Pearson correlation coefficient ($r$) measures the degree of linear correlation in time between model wind speeds and observational wind speeds. Values range from -1 to 1, with -1 indicating a perfect negative correlation, 1 indicating a perfect positive correlation, and 0 indicating no correlation. In Eq. (3) below, $\bar{v}_{\text{mod}}$ is the mean of the modeled wind speeds and $\bar{v}_{\text{obs}}$ is the mean of the observed wind speeds.

$$r = \frac{\sum_{i=1}^{n}(v_{\text{mod},i}-\bar{v}_{\text{mod}})(v_{\text{obs},i}-\bar{v}_{\text{obs}})}{\sqrt{\sum_{i=1}^{n}(v_{\text{mod},i}-\bar{v}_{\text{mod}})^2 \sum_{i=1}^{n}(v_{\text{obs},i}-\bar{v}_{\text{obs}})^2}} \tag{3}$$

Lastly, overlap coefficients (OVLs) were calculated between the probability density functions for the modeled and observed wind speed distributions, using Eq. (4). Functions were estimated using kernel density estimations, specifying Scott's rule (Scott, 2015) for bandwidth smoothing. Once functions were drawn, OVLs were calculated using the following formula, in which $f_{v_{\text{mod}}}(x)$ is the estimated density function for the model wind speeds and $f_{v_{\text{obs}}}(x)$ is the estimated density function for the observed wind speeds. The result of this calculation yields a value from 0 to 1, in which 0 indicates no overlap and 1 denotes complete overlap between the estimated functions for observations and model wind speeds.

$$\text{OVL} = \int_{-\infty}^{\infty} \left( f_{v_{\text{mod}}}(x), f_{v_{\text{obs}}}(x) \right) \, dx \tag{4}$$

In addition to wind speed evaluations, we also conducted wind direction validations using wind roses. This is important for examining the model's performance in capturing the seasonality of wind direction, as well as for investigating the covariance of wind speed and direction (Wu et al., 2022b). For these wind roses, similarity indices (WRSIs) were also calculated by taking the sum of the minimum frequencies between model and observations for each discrete wind direction bin, using Eq. (5). Here, $f_{d_{\text{mod}}}(i)$ and $f_{d_{\text{obs}}}(i)$ represent the frequency of wind directions for each bin $i$.

$$\text{WRSI} = \sum_{i}^{n} \min\left( f_{d_{\text{mod}}}(i), f_{d_{\text{obs}}}(i) \right) \tag{5}$$

## 2.4 Interannual Variability and Model Uncertainty

This section quantifies the magnitudes of model uncertainty and model interannual variability for simulated wind speeds. By exploring the spatiotemporal patterns of both, we can provide crucial insight into wind energy resource applications. Specifically, the degree of interannual variability, as well as the magnitude of model uncertainty, significantly impact the estimated energy yield of a wind farm, consequently determining the cost of investment capital for new wind projects (Pryor et al., 2018; Jung et al., 2019).





Wind, like many other meteorological variables, has interannual variability, driven by climate oscillations and other long-range temporal patterns. Long-term climate models can reasonably capture this variability, allowing for a comprehensive look at year-to-year fluctuations in wind speeds. Additionally, climate model simulations can generate varying solutions when employing different physics parameterizations and initial conditions (Carvalho et al., 2012, 2014; Li et al., 2021; Wu et al.,

2022a; Hahmann et al., 2015; Draxl et al. 2014). This study investigates these two types of variability and discusses their spatiotemporal magnitudes in the context of wind energy applications.

To quantify model uncertainty due to internal variability and structure uncertainty, statistical bootstrapping was employed on the sixteen (10 internal variability, 6 structure uncertainty) 1-year simulations to generate 500 augmented ensemble members. This was done by randomly selecting data for each hour from one of the sixteen ensembles, ultimately

building an entirely new ensemble with the same spatial and temporal domain. This technique allows for a more comprehensive look at the statistical distribution of data and the underlying variability that drives model uncertainty. Time averages were then performed across the model domain on each of the 500 resampled ensembles to gauge how the degree of model uncertainty is influenced by different timescales; this included monthly, bi-weekly, weekly, and daily averages, as well as daytime (21 UTC) and nighttime (06 UTC) monthly averages.

To represent model uncertainty, 5th and 95th percentiles were taken at different time scale averages (e.g., weekly and biweekly) across the 500 augmented ensembles to determine the upper and lower bounds of temporally averaged wind speeds. Then, the difference between these two percentiles (95th - 5th) served to demonstrate the degree of ensemble spread. These percentiles were calculated for every grid point and at each timescale average to reveal spatiotemporal patterns present for model uncertainty. Interannual variability was calculated by taking the same timescale averages, then computing percentiles

across the 20 years of ADDA-v2's full temporal domain.

## 3 Results

### 3.1 Hub-Height Wind speed and Wind Direction Validations

We start with a model validation for wind speeds at hub-heights (Section 2.2) over the 26 locations (Fig. 1) to assess ADDA-

v2's utility for wind energy applications. We used several metrics and statistics to quantify model performance, including probability density functions (PDFs), seasonally averaged wind speed diurnal cycles, wind roses, time-scale dependent RMSE and correlation bar charts. For each figure, locations from the different geographies listed in Table 2 were chosen to assess ADDA-v2's performance in different regions; where possible, at least one figure representing each geographic characteristic was displayed.






### 3.1.1 Probability Density Functions

PDFs effectively compare data distributions without considering the time dimension, aiming to visualize any biases between model and observation. Across the 26 hub-height locations, ADDA-v2's PDFs had an average OVL of 0.85 with the observational PDFs, while ERA5's PDFs had an average OVL of 0.78. Similarities between ADDA-v2 and ERA5 distributions

and observed wind speeds were spatially variable, with ADDA-v2 performing better than ERA5 for 18 of the 26 sites considered. Overall, ADDA-v2 performed well with East Coast CONUS locations, seeing very high OVLs for locations such as Fowey Rock (0.95), Florida, East Point, Florida (0.92), Edith Hammock, Alabama (0.93, Fig. 2i), and Cape Henry, Virginia (0.85). ERA5 also did well for these locations (0.77, 0.95, 0.96, and 0.82 respectively), struggling more just for the Fowey Rock location. For the West Coast of the CONUS, ADDA-v2 and ERA5 on average performed similarly. Megler, Washington,

Los Angeles Pier J (Fig. 2a), California, and Martinez, California (Fig. 2b) saw OVLs of 0.82, 0.9, and 0.79 for ADDA-v2 and 0.94, 0.85, and 0.87 for ERA5 respectively.

Across the Plains region, ADDA-v2 was able to modestly outperform ERA5. The average OVLs for ADDA-v2 across the nine locations was 0.86, while ERA5 saw an average OVL of 0.79. For many of the central U.S. locations, ADDA-v2 wind speed distributions were very close to that of observations, namely Site A, Kansas (Fig. 2e), Site A, Arkansas (Fig. 2f),

and SGP, Oklahoma. ADDA-v2's higher resolution was able to capture the finer scale wind speed patterns in mountainous regions, significantly outperforming ERA5. ADDA-v2 OVLs for the two mountainous regions considered (the Cascades and the Rockies) were 0.91 and 0.87, while ERA5's OVLs were considerably lower at 0.64 and 0.75 (Fig. 2c, d). Additionally, ADDA-v2 outperformed ERA5 for the single Great Lakes location (Fig. 2h), with an OVL of 0.93 compared to 0.82.

There were a couple locations where both datasets struggled to capture the hub-height wind speed distribution. For

example, both ADDA-v2 and ERA5 had low OVLs for the southeast location (Fig. 2j), Spiderweb, South Carolina. ADDA-v2 demonstrated a strong overestimation and saw its minimum OVL of 0.54, while ERA5 has a notably better, but still relatively low OVL of 0.77. As discussed in (Section 3.1), ADDA-v2's positive bias can be partly attributed to the land surface model (LSM) used for these simulations, as well as the positive bias inherited by ERA5. Both datasets also struggled with the hub-height wind speeds at Brookhaven, New York, with OVLs at 0.63 (ADDA-v2) and 0.56 (ERA5). However, the

overestimations seen for this location by both datasets may be attributed to its unique geographic position; it is located on Long Island, New York, equidistant from Long Island Sound and the Atlantic Ocean, where land sea interactions on either side may incite complexities in the local wind patterns. Considering regions outside of the CONUS, ADDA-v2 performed very well across the four Alaska locations, with an average OVL of 0.88, while ERA5 struggled more, with an average OVL of 0.70 (Fig. 2k). ERA5's coarser resolution can contribute to these errors, especially across Alaska, where complex topography

incites stark spatial changes in wind patterns. Specifically, for the Five Fingers location, on the coast of the Kotzbue Sound, ADDA-v2 sees an OVL of 0.93 while ERA5 sees an OVL of 0.77 (Fig. 2k). For the San Juan, Puerto Rico location, ADDA-v2 and ERA5 saw decent performance in capturing wind speed patterns, although ERA5 did demonstrate slight improvement with an OVL of 0.78 compared to ADDA-v2's 0.71.

WIND ENERGY SCIENCE DISCUSSIONS

**Figure 2.** Probability density functions (PDFs) of ADDA-v2 and ERA5 simulated wind speeds alongside observations over Los Angeles Pier J, California (a), Martinez, California (b), Wasco, Oregon (c), NWTC, Colorado (d), Site A, Kansas (e), Site A, Arkansas (f) Site A, Minnesota (g), Dunkirk, New York (h), Edith Hammock, Alabama (i), Spiderweb, South Carolina (j), Five Fingers, Alaska (k), and San Juan, Puerto Rico.





### 3.1.2 Diurnal Cycles

While PDFs are useful in understanding the overall distribution of wind speeds, it is important to validate temporal accuracy of model simulated wind speeds. It is particularly crucial to understand how well the model captures diurnal variability of wind, especially when planning hybrid renewable energy assessments. Therefore, seasonally averaged wind speed diurnal cycles are considered in this analysis for each hub-height location to evaluate how well ADDA-v2 captures intraday wind speed patterns. Ten-meter wind speeds were also included for some of these locations because they have more pronounced diurnal patterns. Pearson's $r$ and RMSE values are used to validate model diurnal cycles against in-situ observations.

Across all locations (Fig. 1), ADDA-v2's diurnal wind speed patterns had an average Pearson's $r$ of 0.67 with observations, while ERA5's average was considerably lower, at approximately $r = 0.35$. Similarly, ADDA-v2 had a lower average RMSE of 1.02 m s$^{-1}$ compared to the 1.36 m s$^{-1}$ RMSE of ERA5. Both datasets saw improved performance when there was a strong diurnal signature in wind speed magnitudes. This was especially the case for southern locations, especially with coastal geographies, where the greater surface heating at lower latitudes modulates diurnal wind speed patterns more significantly (Elliott et al. 2004). For East Coast locations like East Point, Florida, Fowey Rock, Florida, and Edith Hammock, Alabama, Pearson's $r$ were at or above 0.85 for ADDA-v2. ERA5 Pearson's $r$ were also high overall, but the dataset struggled with Fowey Rock, with $r = 0.51$ (Fig. 3b). Across all East Coast locations, ADDA-v2 had an average Pearson's $r$ of 0.72 and an average RMSE of 1.19 m s$^{-1}$ while ERA5 saw a worse Pearson's $r$ (0.61), but a comparable RMSE (1.14 m s$^{-1}$). This trend was generally observed for the West Coast locations as well, in which southern regions had a clear diurnal wind pattern, namely for Los Angeles Pier J, California (Fig. 3a). Overall, ADDA-v2 performed better for the wind speed diurnal pattern for the West Coast region with an average Pearson's $r$ of 0.74 compared to ERA5's 0.64. However, ADDA-v2 did tend to overestimate wind speeds for Martinez, California and Wasco, Oregon, leading to higher RMSE values compared to ERA5.

For more inland regions, namely locations with mountainous or plains geographies, ADDA-v2 performed much better than ERA5 in most statistical metrics considered. Correlation coefficients for plain-like geographies, on average, were $r = 0.76$ for ADDA-v2 and $r = 0.27$ for ERA5. For example, ADDA-v2 excelled at capturing intraday wind patterns across the Great Plains locations, such as SGP, Oklahoma, Site A, Texas, and Site A, Kansas, with $r = 0.83$, $r = 0.91$, and $r = 0.89$ respectively. This was especially the case during the warmer months, when wind speeds had notable fluctuations during the day. RMSEs generally reflected this trend as well, with average RMSEs for ADDA-v2 at 1.01 m s$^{-1}$ and 1.51 m s$^{-1}$ for ERA5. For mountainous regions, both ADDA-v2 and ERA5 struggled significantly to capture diurnal wind speed patterns (Fig. 3c), with an average Pearson's $r$ of 0.32 and 0.24 respectively. The high elevations of these locations have more complex responses to




### Red Dog Dock, AK (2007-2020) 13m Windspeeds

(d)

### San Juan, PR (2001-2020) 10m Windspeeds

(e)

**Figure 3.** Seasonally averaged diurnal wind speeds (summer, winter) for Los Angeles Pier J, California (a), Fowey Rock, Florida (b), Wasco, Oregon (c), Red Dog Dock, Alaska (d), and San Juan, Puerto Rico (e).

diurnal changes in solar heating and thus do not have very clear wind speed patterns throughout the day, especially during the winter (Fig. 3c). ADDA-v2 did outperform ERA5 in the mountainous locations for RMSEs though, with an RMSE value of 1.21 m s$^{-1}$ compared to ERA5's 2.20 m s$^{-1}$.

For the three other regions, Great Lakes, Alaska, and the Caribbean, ADDA-v2 performed modestly better overall than ERA5 in both diurnal correlations and RMSEs. For the Great Lakes location, Dunkirk, New York, ADDA-v2 saw a Pearson's $r$ of 0.82 while ERA5 saw a negative correlation ($r = -0.33$). Their RMSEs were comparable though, at 0.67 m s$^{-1}$ and 0.63 m s$^{-1}$ respectively. Across the four Alaska locations, both datasets struggled to capture the diurnal pattern, with average Pearson's $r$ of 0.46 for ADDA-v2 and 0.12 for ERA5. Diurnal patterns for wind speeds in Alaska, especially for the winter, are mostly nonexistent (Fig. 3d), contributing to these lower correlation values. ADDA-v2 did see much lower RMSE





values, at 0.69 m s⁻¹ compared to ERA5, at 1.64 m s⁻¹. Lastly, for San Juan, Puerto Rico, both datasets were able to capture the dramatic diurnal wind speed pattern observed (Fig. 3e).  However, ADDA-v2 was much more precise, with a correlation coefficient of 0.95 compared to ERA5's correlation of 0.62. ADDA-v2 also had a lower RMSE, at 0.62 m s⁻¹ compared to that of ERA5 at 1.15 m s⁻¹.


### 3.1.3 Wind Roses

PDFs and diurnal cycles were used to assess model performance for wind speeds, but it is also important to assess model performance for wind direction to indicate whether the model can capture synoptic scale phenomena that drive these seasonal changes in wind direction. This section employs wind roses to visualize seasonal wind direction distributions for each hub-
height location between model and observations. ERA5 is not included in this section because of the challenges surrounding interpolating wind direction to different heights.

Across the 19 locations that had available wind direction data, the average wind rose similarity index (WRSI) between ADDA-v2 and observations was 0.75. WRSIs for all observational sites were above 0.6, indicating that ADDA-v2 was able to reasonably capture the climatological synoptic mechanisms driving seasonal changes in wind directions. Also, no single
geographic region significantly outperformed another, with average WRSIs at 0.66, 0.77, 0.78, 0.75, and 0.75 for the west coast, mountains, plains, east coast, and Alaska respectively. The single locations for the Great Lakes and the Caribbean saw relatively high WRSIs of 0.83 and 0.88.

A maximum WRSI of 0.90 was seen for the NWTC, Colorado location (Fig. 4b) where ADDA-v2 was able to accurately capture the predominantly west winds in the fall, winter, and spring, generated by mid-latitude cyclones and the
more mesoscale chinook winds that occur on the leeward sides of mountain ranges (Lackman, 2011; Markowsi & Richardson, 2010). It is noteworthy that ADDA-v2 performed the best in a mountainous region, where wind patterns typically exist at fine spatial scales. ADDA-v2 similarly did well for one of the plains locations, Site A, Minnesota (Fig. 4a), with a WRSI of 0.86. Here, ADDA-v2 was able to capture the northwest component of the wind direction in the winter, modulated by mid-latitude cyclones and frontal passages that are typical during the season. In the spring and summer, ADDA-v2 is able to accurately
simulate the more southerly component of wind, generated by mechanisms such as low-level jets and the more dominant influence of the Bermuda High that dominates during summer (Lackmann, 2011).

ADDA-v2 also performed well for more tropical climates, such as for the near-surface wind directions for San Juan, Puerto Rico (WRSI of 0.88, Fig. 4c), where the winds are driven predominantly by the synoptic-scale Tradewinds over the Atlantic. ADDA-v2 was able to accurately capture the strong eastern components of the wind direction over Puerto Rico,
matching observations closely. This was seen for the Fowey Rock, Florida site, where wind direction is also heavily influenced by the easterly trade winds in the tropics.









(c)

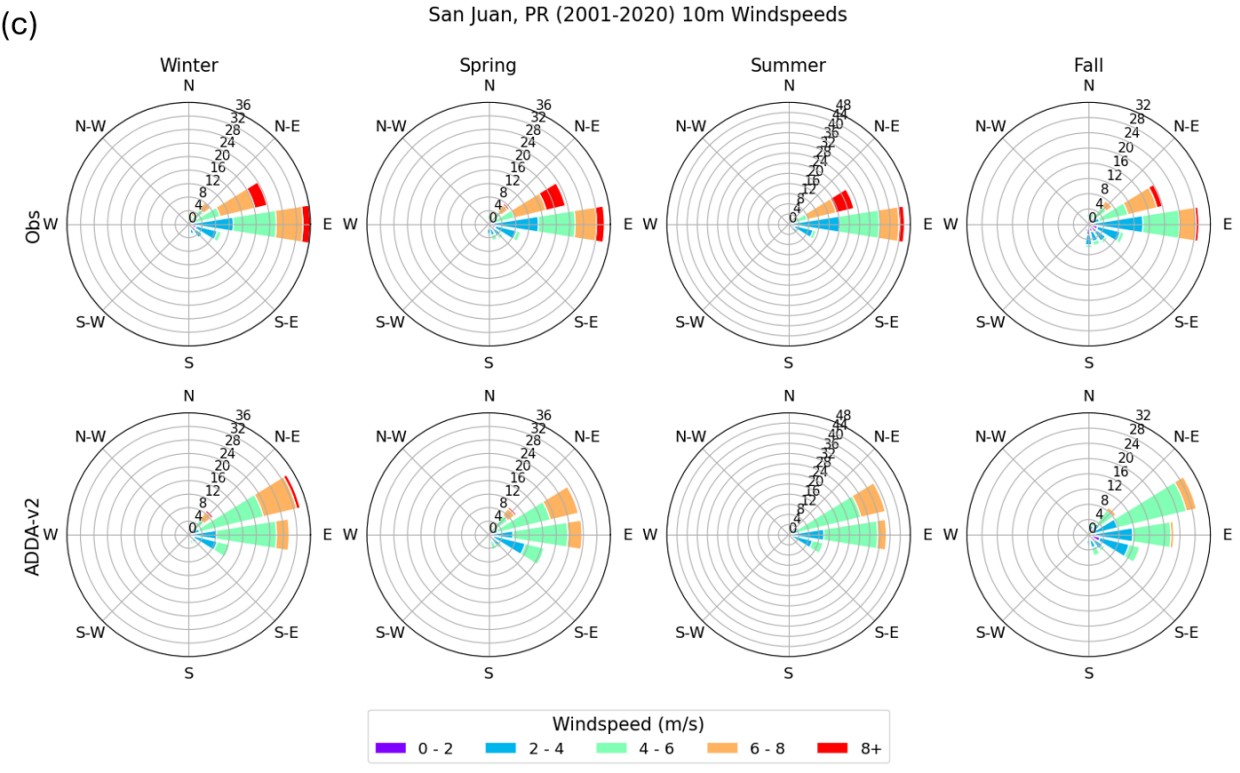


(d)

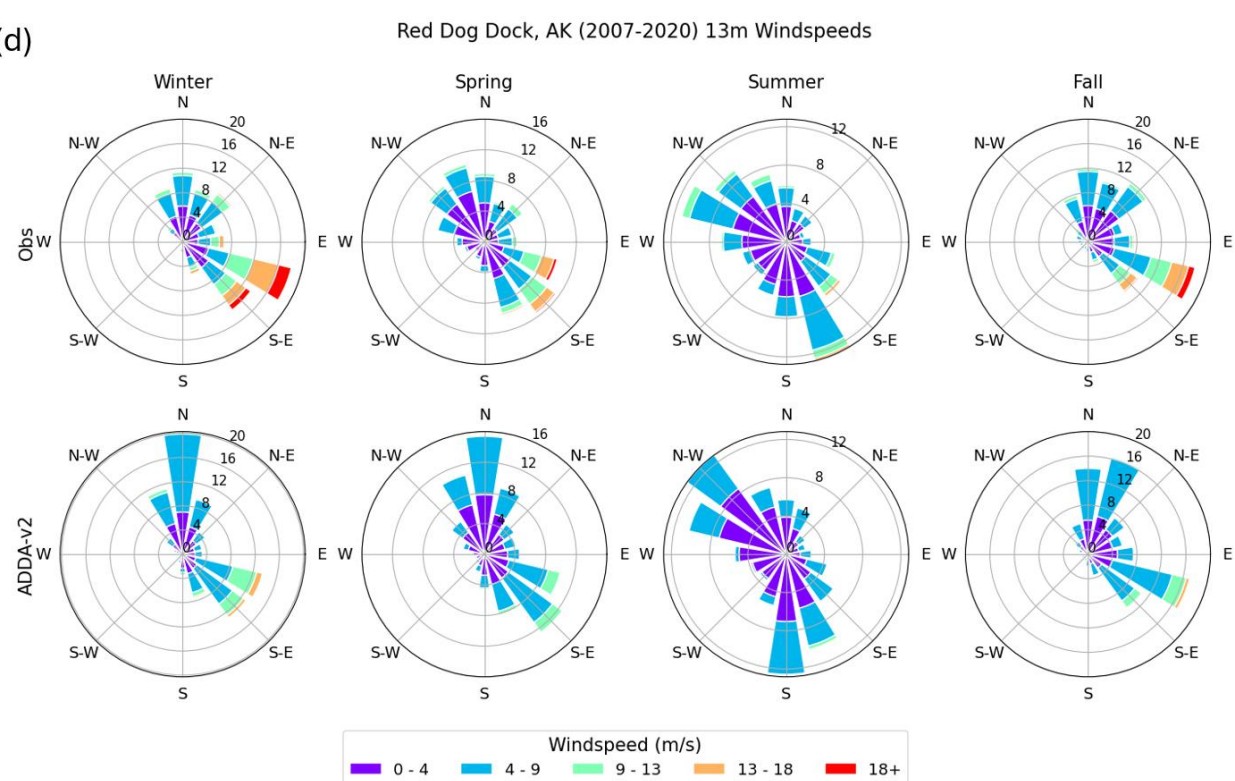





**Figure 4.** Seasonally averaged wind speed and wind direction distributions for Argonne, Illinois (a) and NWTC, Colorado (b). Values on each concentric circle (4, 8, 12, 16) within the wind rose are used to measure the normalized frequency of each wind direction wedge. Windrose wedge positions indicate the direction from which the wind is blowing.


Across the Alaska locations, ADDA-v2 performed moderately well, with wind direction WRSIs at Bligh Reef, Five Fingers, Juneau Dock, and Red Dog Dock (Fig. 4d) at 0.72, 0.81, 0.79, and 0.66. In winter, Alaskan winds are influenced by the extension of the Siberian High, which can bring northerly/northeasterly winds to the northern portions of Alaska during the winter (Fig. 4d). During summer, synoptic scale features are weaker, favoring a heavier influence of mesoscale mechanisms on wind directions patterns. For coastal locations, like Red Dock, Alaska, summer wind directions can be influenced by sea breezes, indicated by the high frequency of southerly flow during the summer (Fig. 4d).

### 3.1.3 Model Performance Across Various Time Scales

The previous evaluations for hub-height wind speeds and wind direction either eliminate the time dimension (PDFs) or look at coarse resolution temporal averages (seasonal diurnal, seasonal wind roses). While there are wind energy applications that require high model accuracy for fine temporal resolutions (minutes, hours), ADDA-v2 is a climate dataset and is therefore not designed for predicting day-to-day weather or weather forecasting. Instead, it is useful for climate scale studies and insightful in understanding climatological patterns for different regions. Thus, we do not expect the model to be able to capture the hour-to-hour variations seen in the observations (Appendix A in Wang et al., 2014). However, this section tests ADDA-v2's capacity to represent wind speeds at different timescales, aiming to demonstrate the timescale in which the model can be useful for wind energy resource assessments.

For almost all hub-height locations analyzed, RMSEs decreased, and correlations increased as the time scale averages became coarser. On average across the 26 locations considered, rRMSEs at the daily, weekly, biweekly, and monthly scale were 46%, 29%, 25%, and 22% respectively, indicating improvement at each transition to a coarser timescale (Fig. 5f). Pearson's $r$ showed a similar trend, at $r = 0.48$, $r = 0.63$, $r = 0.68$, and $r = 0.75$ (Fig. 5f), consistently growing when calculated at increasingly coarse timescales. Intuitively, the daily time scale almost always saw the greatest error between ADDA-v2 and observations, while the monthly time scale performed the best (Fig. 5a-e). Also, the largest error improvement occurred when going from daily averages to weekly averages. RMSEs and correlations improved from weekly to biweekly and again from biweekly to monthly, but not as drastically. For example, this can be seen for the 60m wind speeds at Site A, Arkansas (Fig. 5b), where rRMSEs were at 37% at the daily timescale, before dropping to 20%, 16%, and 13% at the weekly, biweekly, and monthly timescales. Pearson's $r$ also improved from 0.57 at the daily time scale to 0.89 at the monthly timescale. Similarly, Fowey Rock, FL (Fig. 5b) sees a drastic improvement from daily to weekly averaged wind speeds, with rRMSEs dropping from 40% to 23%, and Pearson's $r$ steadily climbing between timescale averages. This trend is seen for the Alaska and Puerto Rico locations as well, with ADDA-v2 struggling to capture day-to-day wind speeds, but performing well at coarser time scales (Fig. 5c, d).



**Table 2.** Statistical metrics for each of the 26 hub-height observational locations.

| Geography | Location | Wind speed OVL | | Wind speed Diurnal Correlation | | Wind speed Diurnal RMSE (m s⁻¹) | | WRSI |
|---|---|---|---|---|---|---|---|---|
| | | *ADDA-v2* | *ERA5* | *ADDA-v2* | *ERA5* | *ADDA-v2* | *ERA5* | *ADDA-v2* |
| W. Coast | Megler, WA | 0.82 | 0.94 | 0.85 | 0.35 | 1.45 | 0.40 | 0.68 |
| | Martinez, CA | 0.79 | 0.87 | 0.39 | 0.64 | 1.64 | 0.86 | 0.61 |
| | Los Angeles Pier J, CA | 0.90 | 0.85 | 0.97 | 0.94 | 0.64 | 0.90 | 0.69 |
| | **Average** | **0.84** | **0.89** | **0.74** | **0.64** | **1.24** | **0.72** | **0.66** |
| Mountain | Wasco, Oregon | 0.91 | 0.64 | 0.69 | 0.4 | 0.78 | 2.53 | 0.63 |
| | NWTC, CO | 0.87 | 0.75 | -0.05 | 0.07 | 1.64 | 1.86 | 0.90 |
| | **Average** | **0.89** | **0.69** | **0.32** | **0.24** | **1.21** | **2.2** | **0.77** |
| Plains | Site A, KS | 0.97 | 0.6 | 0.89 | 0.03 | 0.40 | 2.88 | - |
| | SGP Observatory, OK | 0.90 | 0.89 | 0.83 | 0.89 | 0.52 | 0.76 | 0.79 |
| | Site A, TX | 0.83 | 0.63 | 0.91 | -0.30 | 1.18 | 3.14 | - |
| | Site B, TX | 0.97 | 0.8 | 0.75 | 0.89 | 0.46 | 1.37 | - |
| | Site A, MN | 0.83 | 0.82 | 0.90 | -0.38 | 1.62 | 1.59 | 0.83 |
| | Site A, AR | 0.92 | 0.74 | 0.48 | 0.40 | 0.72 | 1.66 | - |
| | Argonne, IL | 0.76 | 0.89 | 0.64 | 0.55 | 1.15 | 0.35 | 0.82 |
| | Site A, IN | 0.76 | 0.93 | 0.59 | 0.41 | 1.66 | 0.60 | 0.68 |
| | Site A, OH | 0.82 | 0.81 | 0.82 | -0.08 | 1.40 | 1.33 | 0.78 |
| | **Average** | **0.86** | **0.79** | **0.76** | **0.27** | **1.01** | **1.52** | **0.78** |
| Great Lakes | Dunkirk, NY | 0.93 | 0.82 | 0.82 | -0.33 | 0.67 | 0.63 | 0.83 |
| E. Coast | Edith Hammock, AL | 0.93 | 0.96 | 0.86 | 0.93 | 0.59 | 0.31 | 0.72 |
| | Fowey Rock, FL | 0.95 | 0.77 | 0.85 | 0.51 | 0.54 | 1.70 | 0.77 |
| | Spiderweb, SC | 0.54 | 0.77 | 0.62 | 0.23 | 2.09 | 0.84 | - |
| | East Point, FL | 0.92 | 0.95 | 0.92 | 0.95 | 0.68 | 0.31 | 0.70 |
| | Cape Henry, VA | 0.85 | 0.82 | 0.54 | 0.55 | 0.85 | 0.77 | 0.80 |
| | Brookhaven, NY | 0.63 | 0.56 | 0.51 | 0.49 | 2.36 | 2.88 | - |
| | **Average** | **0.80** | **0.81** | **0.72** | **0.61** | **1.19** | **1.14** | **0.75** |
| Alaska | Red Dog Dock, AK | 0.85 | 0.69 | 0.57 | -0.11 | 0.70 | 1.20 | 0.66 |
| | Bligh Reef, AK | 0.90 | 0.86 | 0.39 | 0.25 | 0.55 | 0.96 | 0.72 |
| | Juneau Dock, AK | 0.83 | 0.47 | 0.48 | 0.33 | 0.90 | 2.90 | 0.79 |
| | Five Fingers, AK | 0.93 | 0.77 | 0.40 | 0.01 | 0.60 | 1.50 | 0.81 |
| | **Average** | 0.88 | 0.70 | 0.47 | 0.12 | 0.69 | 1.64 | 0.75 |
| Caribbean | San Juan, PR | 0.71 | 0.78 | 0.95 | 0.62 | 0.62 | 1.15 | 0.88 |
| **All** | **Average** | **0.85** | **0.78** | **0.67** | **0.35** | **1.02** | **1.36** | **0.75** |

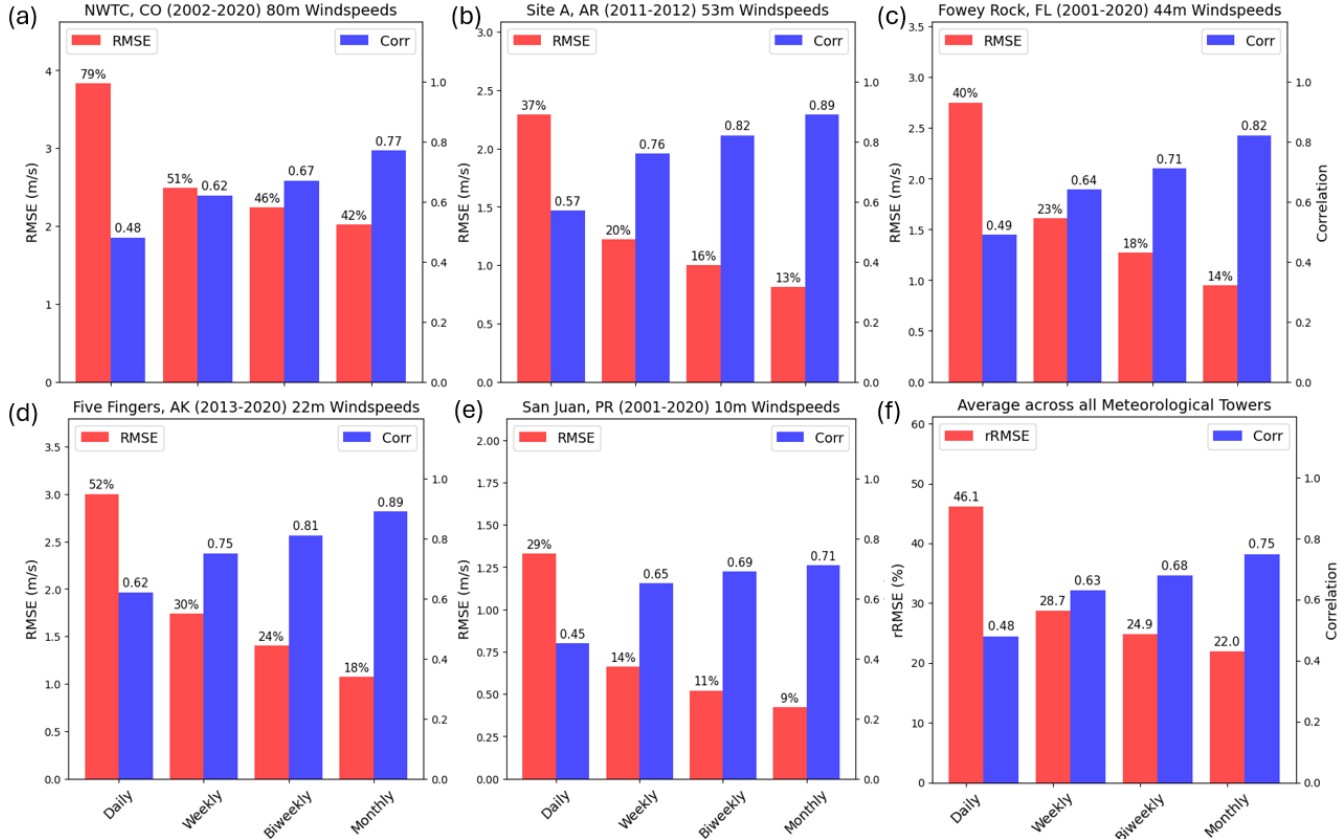

**Figure 5.** ADDA-v2 RMSEs, rRMSEs, and Pearson correlation coefficients at different timescale averages for Site A, Arkansas (b), Fowey Rock, Florida (c), San Juan, Five Fingers, Alaska (d), Puerto Rico (e), along with average metrics across all 26 meteorological towers (f). The number on each bar represents the value for each respective statistic, with time scales becoming coarser from left to right.

As mentioned, the climatic nature of this dataset implies the inability to utilize ADDA-v2 at fine temporal scales. However, ADDA-v2 captures sub-seasonal scales reasonably well, which can give insightful indications for optimal wind farm siting. With ideal siting, higher-resolution models can then be employed to simulate intraday wind patterns to accurately forecast energy production.

## 3.2 Near-surface Wind speed Evaluation

ADDA-v2 near-surface validations were initially performed using wind speed observations taken from 2,000+ ASOS stations across CONUS, Alaska, and Puerto Rico. While the full temporal domain (2001-2020) of ADDA-v2 was used in this analysis, statistics for each ASOS station were dependent on the maximum overlap in data availability between ADDA-v2 and observations. Seasonal means were taken across the available temporal period before calculating RMSE values for each ASOS station.





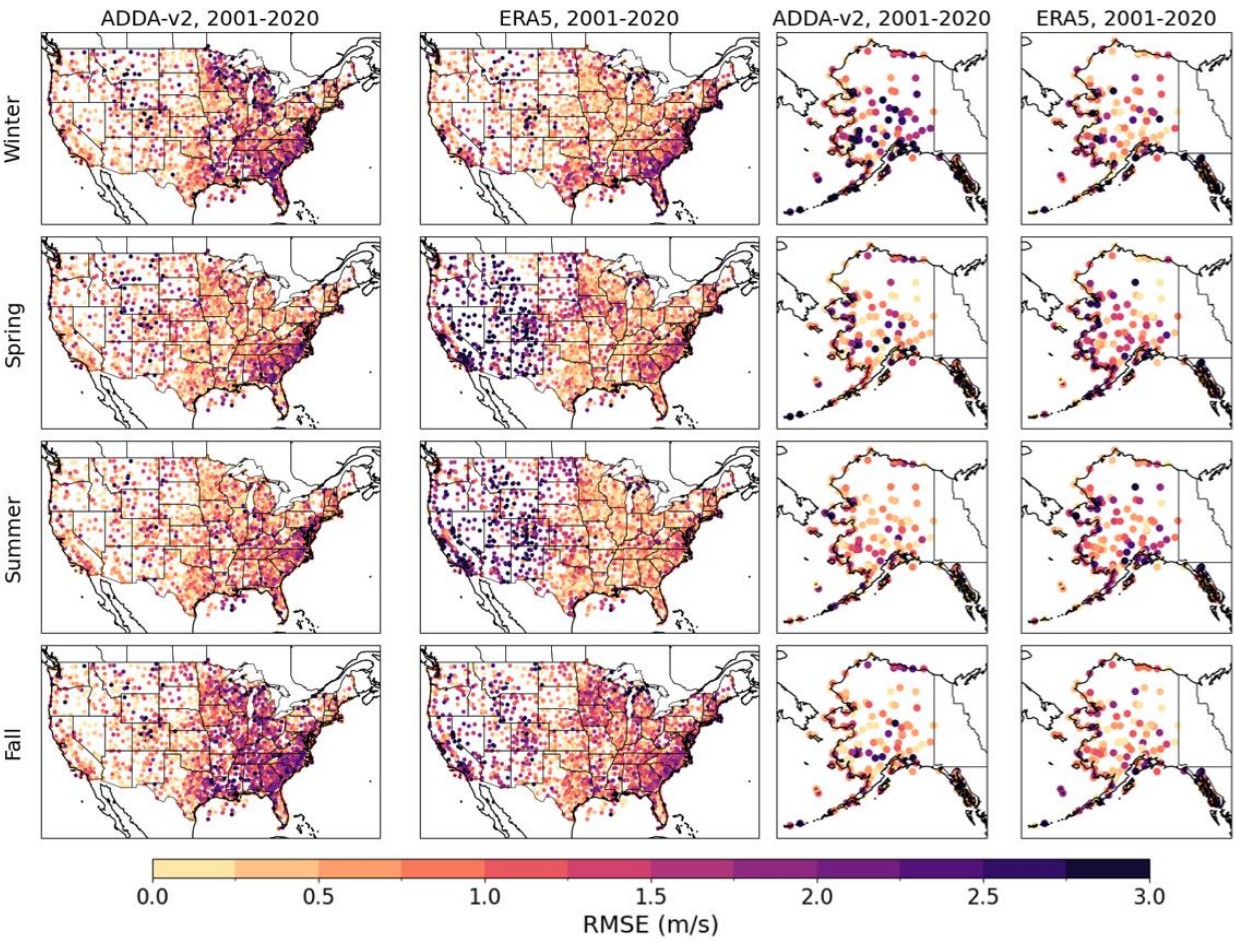

**Figure 6.** ADDA-v2 and ERA5 seasonal RMSEs calculated against 2,000+ ASOS locations across CONUS and Alaska.

ADDA-v2 performs well for the majority of ASOS stations evaluated, with RMSE values falling between 0 m s$^{-1}$ and 1 m s$^{-1}$ across much of the model domain. Spatially, ADDA-v2 accurately captures wind speeds for much of the western portion of CONUS (RMSEs between 0-0.5 m s$^{-1}$), whereas ERA5 struggles significantly, especially in the spring and summer (RMSEs upwards of 3 m s$^{-1}$). This has been documented in past studies (Chen et al., 2024; Wilczak et al., 2024), which highlight ERA5's tendency to underestimate wind speeds in areas of complex terrain (i.e., the Rockies).

For the eastern half of CONUS, both ADDA-v2 and ERA5 show similar spatiotemporal patterns for error magnitudes. Specifically, both datasets demonstrate moderate RMSE values across the Southeast (1-2.5 m s$^{-1}$), most notably during the fall and winter. This systematic error is predominantly attributed to model overestimation during nighttime hours (00 - 12 UTC), when observational wind speeds are very low (0-1 m s$^{-1}$). Interestingly, ADDA-v2 also shows higher RMSE values for the upper Midwest during the fall and winter, when wind speeds are seasonally stronger; this bias is analyzed more in depth in



Section 3.3. For most other regions, namely the central/lower Midwest, Texas, and the Northeast, ADDA-v2 and ERA5 accurately capture seasonal wind speeds, indicated by low RMSE values.

Overall, ADDA-v2 slightly outperforms ERA5 when considering the mean error across all ASOS stations used in this analysis. Across winter, spring, summer, and fall, ADDA-v2 saw average RMSE values at 1.06, 0.87, 0.82, and 1.05 m s$^{-1}$ respectively, with a full-year average of 0.95 m s$^{-1}$. ERA5 saw average RMSE values at 0.96, 1.12, 1.07, and 1.13 respectively, with a full-year average of 1.07 m s$^{-1}$. ADDA-v2 performed best during spring and summer, when wind speed overestimations were reduced in the Southeast. Alternatively, ERA5 performs best during the winter, when the large error over the West is minimized.

When specifically looking at the ASOS stations over Alaska (Fig. 6), ADDA-v2 and ERA5 generally capture coastal wind speeds well, but struggle more in areas with complex topography. For some locations of Alaska's mountainous interior, RMSE values are much higher than surrounding locations (RMSEs greater than 2.5 m s$^{-1}$). Overall, average RMSEs across Alaska for each season were 1.65, 1.08, 0.9, and 0.95 m s$^{-1}$ for ADDA-v2 and 1.14, 1.23, 1.17, and 0.96 m s$^{-1}$ for ERA5. Full-year RMSE averages were almost identical, at 1.14 and 1.13 m s$^{-1}$ for ADDA-v2 and ERA5 respectively. Similarly to CONUS, ADDA-v2 was able to more accurately capture Alaska's wind speeds during the summer and fall but had a notable spike in RMSE magnitudes during the winter, especially for inland locations.

Overall, both ADDA-v2 and ERA5 are able to reasonably capture the seasonal patterns in near-surface wind speeds across CONUS. ADDA-v2 does have a slight edge in performance, but both datasets demonstrate some systematic biases. The strong underestimation of ERA5 wind speeds over the western region of CONUS is noteworthy and could have significant implications to data utilization in that area. Likewise, ADDA-v2 shows an overestimation over the Southeast in the fall and winter, when observed wind speeds are very low, especially during the overnight hours.

### 3.3 Sensitivity of Wind speed Biases to Physics Parameterizations

Most notably, ADDA-v2 sees positive wind speed biases across the Southeast United States, as well as for some parts of the Upper Midwest. This bias is seen for both the near-surface winds and the hub-height winds (Fig. 6, 2e). Primarily, this is attributed by the biases within the forcing data used as boundary conditions to run ADDA-v2 simulations. Depicted in Fig. 6, ERA5 demonstrates relatively higher RMSE values for southeast CONUS, overestimating wind speeds for this region. ERA5 most significantly overestimates wind speeds during overnight hours, when observational wind speeds fall between 0-1 m s$^{-1}$. ADDA-v2 inherits this bias and sees wind speeds in the southeast 1-2 m s$^{-1}$ higher than observations.

This tendency to overestimate wind speeds at night may also be attributed to the model's capacity to respond to atmospheric stability. It has been documented that Noah-YSU (the PBL and LSM schemes used to run ADDA-v2 simulations) has an enhanced performance for wind speeds in unstable conditions but struggles in a very stable atmosphere (Hong et al. 2006, Draxl et al. 2014, Wang and Jin 2014). Thus, the very low wind speeds present during stable conditions may not be accurately captured by models employing these schemes.



To further investigate the implications of model schemes on simulated wind speeds, we validated all structure uncertainty (Section 2.4) ensemble members against observations in regions where ADDA-v2 experiences positive wind speed biases. This allowed us to test whether an alternate model configuration could achieve enhanced performance in areas where ADDA-v2 demonstrated near-surface wind speed overestimations. Various ASOS locations were chosen in areas where ADDA-v2 showed high RMSE values and seasonally averaged diurnal cycles were plotted across the six ensemble members

against observations. Error metrics were calculated and the most accurate ensemble, indicated by the highest correlation coefficient or the lowest RMSE, was noted (Fig. 7). Because the Southeast wind speed overestimations can at least be partly attributed to the inherited bias from the ERA5 forcing data, locations were chosen within the Midwest region to test the performance of different model configurations.

**Figure 7.** Seasonally averaged diurnal cycles for each of the structure uncertainty ensemble members (Section 2.4) against observed wind speeds in regions where ADDA-v2 demonstrated a positive bias.



For each location that demonstrated a positive near-surface wind speed bias, the Noah-MP land surface model outperformed the Noah land surface model, as seen in the diurnal cycles plotted for a Wisconsin ASOS station. (Fig. 7). It is

480 also apparent that the greatest error occurs during the overnight hours (00-12 UTC), in which none of the six ensemble members come close to representing the observed wind speeds. Contrarily, during the daytime hours (12-00 UTC), all ensemble members are able to more accurately capture wind speed magnitudes, although still demonstrating some degree of overestimation. Furthermore, in all but one metric, the MYNN PBL scheme outperformed the YSU PBL scheme. Of the statistical metrics considered for each season, the MYNN PBL scheme almost always showed the lowest RMSE value and the

485 highest correlation coefficient. However, it is important to acknowledge that no individual model configuration was able to solve the positive bias seen for these locations.

Considering the effects that different LSM schemes have on simulated wind speeds, we further analyzed how specific LSM parameterizations drive differences in near-surface winds. One of the most important considerations is the friction velocity, typically denoted by u*, in which a greater magnitude of this variable corresponds to weaker wind speeds. Friction

velocity is essential in accurately representing boundary layer processes and is crucial for accurately simulating wind profiles.

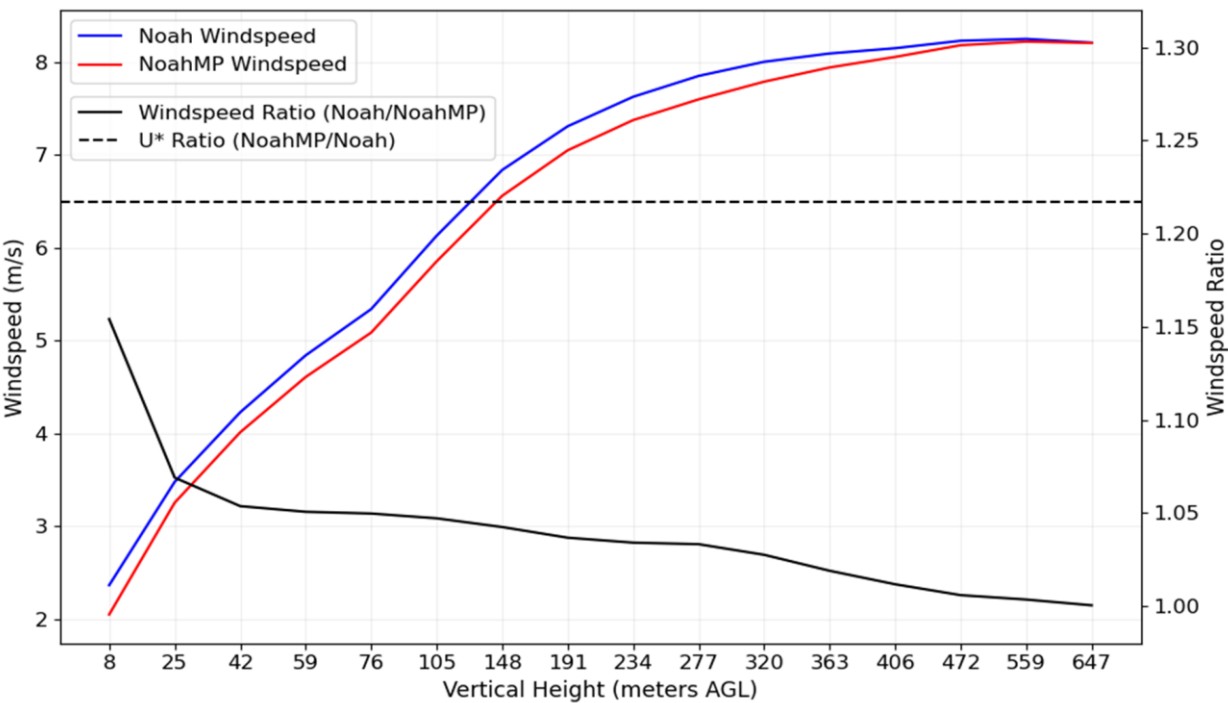

**Figure 8.** Vertical profile of wind for a location in which ADDA-v2 overestimated wind speeds, comparing averaged winds between the
495 2018 simulations using the NoahMP and Noah LSMs. Wind speed profiles correspond to the leftmost y-axis while the ratios of both wind speed and friction velocity use the rightmost y-axis.





However, in different model configurations, this parameterization can vary. We analyzed this between the Noah-MP and Noah LSM schemes and found that friction velocity generally tends to be larger in model simulations that employ Noah-MP (Fig. 8). Then, looking specifically at the locations that saw positive near-surface wind speed biases, it was discovered that Noah-MP showed a notably greater friction velocity when compared to that of the Noah LSM. In some cases, the friction velocity was as much as 20-25% larger in NoahMP than Noah. This has significant implications on wind speeds close to the surface, where greater friction velocities signify weaker winds and can have significant influences on model performance.

As seen in Fig. 7, the NoahMP LSM tended to simulate weaker, and more accurate winds, than its Noah counterpart. This can be partly attributed to the greater magnitude of the friction velocity coefficient parameterized in the NoahMP LSM. This factor was crucial in driving the difference in simulated wind speeds between models employing these different LSMs. Wind speed ratios between NoahMP and Noah, specifically within the first ~10m AGL, were as high as 1.15 (Fig. 8). At greater heights, this ratio decreases as friction has a diminishing influence on momentum fluxes with height and wind speeds get stronger overall. However, it is important to note that while the NoahMP LSM saw improved performance in simulating near-surface winds, it still did not fully resolve the positive bias observed.

## 3.4 Interannual Variability and Model Uncertainty

Interannual variability across ADDA-v2's 20-year temporal period was calculated across the entire spatial domain. Additionally, model uncertainty was quantified by investigating the spread across 500 augmented ensembles, varying in their physics parameterizations (structure uncertainty) and initial conditions (internal variability). Then, the magnitudes and spatiotemporal patterns of each of these variabilities were investigated.

Intuitively, the degree of model uncertainty is significantly influenced by the timescale being analyzed. This can be seen in Fig. 9, in which the magnitude of uncertainty scales inversely with the length of the timescale. The biweekly timescale sees uncertainty values of approximately 0.4-0.7 $m\,s^{-1}$ across much of North America (Fig.9, 10). The weekly timescale sees a notable increase in the uncertainties observed, with most values falling between 0.7-1.1 $m\,s^{-1}$. Lastly, the daily timescale sees the most drastic increase in uncertainty, with many locations across North America seeing values exceed 2.5 $m\,s^{-1}$. This concept is similar to the improved performance of ADDA-v2 at coarser resolution timescales (Section 3.1.4). Also, daytime and nighttime monthly averages were also analyzed to explore the diurnal signatures of model uncertainty. Both yielded comparable uncertainties overall, but nighttime monthly averaged uncertainties were slightly higher than that of daytime monthly averaged uncertainties, typically by about 0.2-0.4 $m\,s^{-1}$. Nighttime uncertainties were noticeably higher in the regions of complex topography, especially over the Rocky Mountains.

It is also important to note the spatial patterns present for model uncertainty across the domain. In regions with more complex topography, model uncertainty tends to be higher. For the biweekly and weekly timescales, the mountainous regions demonstrate higher uncertainty values, by about 0.5 $m\,s^{-1}$, when compared to adjacent regions with simpler topography (Fig. 9, Fig. 10a, b). Complex topography introduces more unpredictable interactions between the physical mechanisms that drive







**Figure 9.** Model uncertainty at different timescale averages (daily, weekly and biweekly), represented by the difference between the 95th and 5th percentiles of the wind speed distributions.

near-surface wind (Wu et al., 2022a; Helbig et al., 2017). Thus, small changes in model initial conditions or parameterizations can influence these mechanisms and cause significant variability within the simulated wind. It is also interesting to note though that large lake features also observed high degrees of model uncertainty, specifically during the summer months.

In the context of wind energy applications, model uncertainty is integral when mapping ideal locations for wind farm siting. However, it needs to be paired alongside spatiotemporal patterns of interannual variability to understand the full scope





**Figure 10.** Biweekly averaged interannual variability and model uncertainty for one winter month (January) and one summer month (July).
Uncertainty and interannual variability were taken as the difference between the 95th and 5th percentiles of the wind speed distributions at
each timescale average.





of wind resource reliability and potential risks associated with long-term power generation. Ideally, both model uncertainty and interannual variability need to be low for optimal and consistent power generation. As seen in Fig. 10a, the magnitude scale of interannual variability and model uncertainty is very different. For example, weekly averaged 100 m wind speeds can fluctuate as much as 6-7 m s$^{-1}$ between years, especially for the winter months, when highly variable synoptic-scale features strongly influence wind patterns. Alternatively, model uncertainty is typically in the range of about 0.5-1.5 m s$^{-1}$ for the same season and time-scale average.

The summer season shows a notable decrease in interannual variability, with typical magnitudes ranging from 3-4 m s$^{-1}$, most likely attributed to the more consistent synoptic patterns present during summer. Model uncertainty during the summer, however, shows similar magnitudes to that of winter and has a relatively constant spatial pattern (Fig. 10b). This was seen at other timescales as well (i.e. biweekly/monthly), in which the magnitude and spatial patterns of model uncertainty remained relatively consistent between seasons. As aforementioned, topographically complex regions saw much higher magnitudes for model uncertainty, and the spatial patterns of interannual variability matched this trend as well. For both winter and summer, regions of complex terrain (i.e., the Rocky Mountains) saw higher magnitudes of interannual variability than surrounding regions.

Altogether, both components of uncertainty, interannual variability and model uncertainty, are integral considerations for wind resource assessments. This data can be leveraged to identify key regions that have an optimal combination of moderately strong wind speeds and relatively low model uncertainty and interannual variability. Ultimately, this will maximize energy output potential for optimally sited wind farms and minimize the risk of unpredictable extreme weather events that can affect wind energy infrastructure.

## 4 Discussion and Concluding Remarks

The validation of the Argonne Downscaled Data Archive Version 2 (ADDA-v2) dataset presented in this study underscores its utility in wind resource assessments and climatological applications. This section synthesizes the key findings and compares the performance of ADDA-v2 with ERA5, highlighting ADDA-v2's added value to its coarser resolution forcing data.

ADDA-v2 demonstrated significant advantages over ERA5 in capturing fine-scale wind variability across diverse geographies. The dataset performed particularly well in regions with complex terrain, such as the Rocky Mountains and Alaska, where high-resolution modeling captured localized wind phenomena more effectively. Specifically, ADDA-v2 outperformed ERA5 in hub-height wind speed distribution evaluations, with an average OVL of 0.85 compared to ERA5's 0.78. Temporal validation further emphasized ADDA-v2's improved capabilities: ADDA-v2 exhibited strong correlations with observed diurnal cycles, indicated by an average correlation of 0.67 across all locations analyzed compared to ERA5's 0.35. This is especially critical when assessing the consistency in wind power generation throughout the day, with potential implications



for hybrid style energy generation. Additionally, ADDA-v2's ability to reduce errors at coarser temporal scales (e.g., weekly and monthly averages) reinforces its applicability for long-term climatological studies and resource planning.

However, challenges remain, particularly in regions where both ADDA-v2 and ERA5 struggled, such as the Southeast United States and areas influenced by stable atmospheric conditions. These limitations highlight the need for targeted improvements in parameterizations to address specific biases. But as this analysis found, none of the model configurations tested across the six structure uncertainty ensembles were able to resolve the biases, indicating partial attributions to both the inherited bias from the ERA5 forcing data as well as the model's systematic bias, resulting from incomplete parameterizations,

such as the specific PBL and LSM parameterizations and limitations in capturing complex interactions within the model's climate system. Future research can look more in depth at the specific mechanisms within these parameterizations to understand why they are unable to capture certain wind patterns. Additionally, while this validation focused more on inland regions, future analysis may expand the validation to offshore locations, testing ADDA-v2's performance over coastal and oceanic locations. (Research has been conducted exploring ADDA-v2's capability at capturing coastal winds in Sheridan et

al. (2024).

Ultimately, the validation presented here, coupled with the analysis of model uncertainty and interannual variability can provide the framework for useful applications of the ADDA-v2 dataset. By providing detailed insights into wind resource variability, ADDA-v2 enables more informed decisions in renewable energy planning, risk assessment, and climatological studies.


**Code and Data Availability**

All datasets used in this study are freely available, except for the selected proprietary hub-height data. ERA5 reanalysis data is accessible through Climate Data Store: https://cds.climate.copernicus.eu/. ADDA-V2 data is located on the ALCF high-

performance storage system and is available upon request. A subset of the ADDA-v2 dataset is hosted by the National Renewable Energy Laboratory, providing access to hub-height wind speed data (https://developer.nrel.gov/docs/wind/wind-toolkit/wtk-led-climate-v1-0-0-download/). The public in-situ data can be found on Data Archive and Portal (DAP) Platform (https://a2e.energy.gov/data) and the IEM Mesonet (https://mesonet.agron.iastate.edu/ASOS/). Data processing scripts were written in python and can be made available upon request.


**Author Contributions**

Analysis, software development, and manuscript preparation were performed by KP and JW. Model uncertainty quantification analysis was initiated by JF. Observational data management and processing was performed by LS. The model dataset used in this study was developed by CJ, GS, RK, and CD and post-processed by EY, AP, and AK at hub-heights. All authors were

involved in research conceptualization, discussion, and manuscript edits.





**Competing Interests**

The contact author declares that none of the authors have competing interests.

**Acknowledgement**

This work has been supported by the Laboratory Directed Research and Development (LDRD) Program at Argonne National Laboratory through the U.S. Department of Energy (DOE) contract DE-AC02-06CH11357 and by the Wind Energy Technologies Office (WETO) of the U.S. Department of Energy Office of Energy Efficiency and Renewable Energy. Computational resources for creating the datasets and for data analysis were provided by the Argonne Leadership Computing
Facilities (ALCF) including Theta, and Polaris, and National Renewable Energy Laboratory (NREL)'s High Performance Computers including Eagle and Kestrel. We thank Dr. Larry Berg from Pacific Northwest National Laboratory for his insight into the ensemble design and the analysis at the early stage of this work.

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
