# Peer review of "Evaluation of a High-Resolution Regional Climate Simulation for Surface and Hub-height Wind Climatology over North America"

_Wind Energy Science, 2025_

## Referee Comment (RC1)

**Overall feedback**

This manuscript provides an extensive evaluation of the ADDA-v2 climatological wind dataset for North America. The manuscript text is logically structured and well-written and the figures are clear. A large set of measurement stations and different aspects of the model performance are considered over a range of timescales. The analysis goes into sufficient depth and an uncertainty analysis is included. It is also valuable that a comparison is done to the widely used dataset used to drive the model runs, i.e. ERA5 to see where the downscaled dataset significantly improves on the driving data. However, some choices or sections in the manuscript would benefit from changes or additions, which I will detail below.

**General comments**

1. It would be valuable if the findings of this evaluation (and the ADDA-v2 dataset itself) would be compared to other evaluation studies (and other datasets), preferably for the region. See also specific comment nr. 2. Hence, some references should be added in the discussion section (or somewhere else) to better situate this wind product in relation to other datasets and also how the evaluation/performance differs from other studies.

**Specific comments**

1. Page 2, line 40: I recommend adding that fine-scale surface variations (topography, land cover, …) are also better represented.

2. Page 2, Line 60: Considering that this wind dataset of Draxl et al. (2015a,b) exists and has an even higher spatial resolution, could you add somewhere in the manuscript why this new data product ADDA-v2 is perhaps a better choice going forward?

3. Page 3, line 72: 20 years is indeed a substantial length, but climate variability at 20-year timescales still exist – it could be valuable to mention somewhere in the manuscript how the ADDA-v2 dataset can be supplemented to account for this in wind resource assessments.

4. Page 4, line 98: Could you provide motivation for the choice of two months of spin-up? It is rather limited for the soil component as says a publication of Jerez et al (2020) [https://doi.org/10.1029/2019MS001945], definitely if simulations start in the cold months.

5. Page 4, line 122: On such a large domain, why is no spectral nudging employed for small wavenumbers? Has it been validated at some point that the properties of synoptic systems are indeed adequately reproduced? It would be good to comment on this.

6. Page 5, table 1: were there any model options activated to account for subgrid-scale orography (e.g. topo_wind for YSU PBL scheme or the GWDO scheme) – would be good to mention this.

7. Page 5: table 1: can you motivate the choice for 49 vertical levels? Low-level winds are usually quite sensitive to this choice. Perhaps sensitivity tests were conducted?

8. Page 5, line 138: are the observations also corrected for mast flow distortions? Perhaps good to comment on this and the implication on observational uncertainty.

9. Page 11, line 267: "As discussed in (Section 3.1)" should be corrected.

10. Page 11, line 277: I would not use "improvement" here.

11. Page 13, line 293: So is this the r between the seasonally-averaged values? Or is it the seasonally-averaged value of daily r values? Sometimes in the manuscript this is not very clear.

12. Page 19, section 3.1.3 (wind roses): I agree that a good approximation of the wind rose is a first indication that synoptic winds are well captured. However, the manuscript would benefit from any additional analysis (or references to related studies) which looks at this in more detail. I mainly say this because the domain is very large and no nudging is used. If this cannot be provided, it would be good to mention that this has not been inspected in full detail.

13. Page 19, line 377: section index 3.1.3 is repeated here.. should be 3.1.4 I think?

14. Page 21, line 409: your prior analysis of seasonally-average diurnal cycles shows that statistically (not for specific days) the diurnal cycle is captured well. Here you seem to suggest to the reader not to use the sub-daily information of ADDA-v2. Is this not too critical? If you would feed the diurnal cycles of 500 winter days from ADDA-v2 to a wake model to check performance of a wind farm, I don't feel like the output would not be trustworthy. So maybe rephrase this a bit or provide additional clarification.

15. Page 22, figure 6: As surface wind speeds are usually well below 10 ms-1 on average, RMSE does not give a good idea of how good the agreement actually is. Could you use the rRMSE here?

16. Page 22, figure 6: In the introduction you mention that wind datasets are important for risk assessments of high winds. Yet, for this surface wind speed evaluation, you focus on seasonal RMSE's. Why not also look at extremes? Is there perhaps a reason why this dataset is not intended for looking at extreme winds? Logically, for wind energy purposes one would be interested in winds higher up, so I'm curious what the surface wind speed output is for. You could add an analysis, or clarify why the presented analysis is sufficient.

17. Page 24, figure 7: These indicators '7-2', '5-1' – where do these come from? These should be explained somewhere.

18. Page 25, line 498: Instead of saying that high friction velocities correspond to weaker winds, could you explain a bit better to the reader why this is the case?

19. Page 26, section 3.4: An interesting addition. However, I am very interested to see the importance of the two components of model uncertainty: initialization and physics choices. I would expect that the lateral forcing would lead the initialization to not matter too much on annual statistics compared to physics parametrization choices. Please add this.

20. Page 28, figure 10: Is the inter-annual variability over a specific 2-week period something that wind farm operators are concerned about? As expected, this variability is very large. I think that the inter-annual variability on seasonal timescales might be good to add as well: e.g. how good or how bad a winter period can be in terms of wind resource over the lifetime of a wind farm. You could motivate your choice or add also the seasonal timescale to this analysis.

21. Page 29, line 553: Is the lower inter-annual variability in summer not simply a consequence of lower wind speeds? Relatively speaking, the variability could be as large as for winter or even larger.

22. Page 30, line 601: I get a "404 not found" when pasting the link to reach the hub height wind data. Please make sure that a reliable pointer is available to access the data and that some documentation is available there. Perhaps also for the full ADDA-v2 data: include an e-mail / link where to request access.

---

## Referee Comment (RC2)

**General feedback**

Overall, this manuscript is well written, well-structured and appears carefully worked through with nice looking figures. The work describes a new 4km, 20y mesoscale dataset covering North America with extensive validation using met towers and surface stations and ensemble analysis for a selected period.

Downscaling of global reanalysis models using mesoscale models like WRF is well covered in the literature as well as the improvements it provides relative to the global models. Hence, the novelty of the approach in this manuscript may be disputed given that it has poorer resolution compared to the previous work of e.g. Draxl et al. (2015). However, the open access to the large dataset and the extensive validation effort including ensemble analysis justifies the publication.

**General comments**

In general, I would like to question if the selected validation metrics for wind speed (r, RMSE, rRMSE, OVL) provide sufficient complementary insight. In my view, these metrics overlap too much in what they measure and none of them allow for distinction between systematic errors (biases) and fluctuating errors. I suggest including a simple metric like mean (bias) error to cover this important aspect and re-reconsider if each of the other metrics contribute enough additional insight to remain in the paper. A metric should be included only if characteristic error structures can be inferred from it – to move beyond being merely descriptive.

I suggest reducing the mostly summarising parts (section 3) with long descriptions and lists of numbers in the text. Please also consider additional summary table(s) for better overview and readability.

The paper should include consideration/discussion of the effect of not accounting microscale effects. A 4km model effectively resolves scales from 20-30km and up. How is this expected to affect presented results, when validating the model against measurements that include significant effects on finer scales, which may be very strong at 10m agl.?

Argumentation that the selected ensemble runs represent model uncertainty should be strengthened, this currently is an implied assumption. Does the spread across the selected and boot-strapped ensamples really represent actual model uncertainty?

The limitations and uncertainty of the observations used in the validation should be discussed either in section 2.2 or section 4.

**Some detailed comments**

Page 2, line 63:     It should be mentioned here that ERA5 is initial/boundary model in addition to the info in table 1, on page 5.

Page 10, line 223:     Explain "internal variability" and "structure uncertainty" in more detail, and why 10 and 6 ensemble members , respectively, was decided upon.

Page 14-15, fig. 3:     A legend is missing for the plots.

Page 15, line 340:     Interpolation in wind direction simply requires conversion of wind direction to components which may be interpolated similar to the wind speeds, and then converted back to wind directions.

Page 25, line 489:     Friction velocity is denoted using $u_*$ and not u*.

Page 26, line 503:     Explain why "high friction velocities correspond to weaker winds"

---

## Author Comment (AC1)

**Overall feedback**

This manuscript provides an extensive evaluation of the ADDA-v2 climatological wind dataset for North America. The manuscript text is logically structured and well-written and the figures are clear. A large set of measurement stations and different aspects of the model performance are considered over a range of timescales. The analysis goes into sufficient depth and an uncertainty analysis is included. It is also valuable that a comparison is done to the widely used dataset used to drive the model runs, i.e. ERA5 to see where the downscaled dataset significantly improves on the driving data. However, some choices or sections in the manuscript would benefit from changes or additions, which I will detail below.

Thank you for the insightful comments which we find valuable in improving our manuscript.  We have responded to each of the following general and specific comments (in blue font) and will make appropriate adjustments during the revision process.

**General comments**

1. It would be valuable if the findings of this evaluation (and the ADDA-v2 dataset itself) would be compared to other evaluation studies (and other datasets), preferably for the region. See also specific comment nr. 2. Hence, some references should be added in the discussion section (or somewhere else) to better situate this wind product in relation to other datasets and also how the evaluation/performance differs from other studies.

We agree and plan to contextualize the added value of ADDA-v2 to existing datasets that cover a similar region, including the wind dataset discussed by Draxl et al. (2015a, b). Such studies were mentioned in the Introduction (Draxl et al., 2015a, b; Gensini et al., 2023; Liu et al., 2017; Rasmussen et al., 2024) but will be readdressed in the *Discussion* to situate ADDA-v2 in relation to these other high-resolution datasets. We will also add a discussion about the relative performance of ADDA-v2 compared to other datasets, as studied in our collaborative work by Sheridan et al. 2024.

**Specific comments**

1. Page 2, line 40: I recommend adding that fine-scale surface variations (topography, land cover, …) are also better represented.

   Agree. Will do.

2. Page 2, Line 60: Considering that this wind dataset of Draxl et al. (2015a,b) exists and has an even higher spatial resolution, could you add somewhere in the

manuscript why this new data product ADDA-v2 is perhaps a better choice going forward?

Referring to the response to the general comment, we plan to include a component to the discussion that addresses existing datasets and outlines the added merit of ADDA-v2 in climatological wind studies or wind-resource assessments. We plan to highlight the more extensive temporal coverage of ADDA-v2 as well as the unique spatial domain that spans into regions typically outside the domain of other datasets, such as Alaska, Mexico, and Caribbean islands. Additionally, the ensemble analysis is useful for model uncertainty quantification; and the extensive temporal coverage can help examine inter-annual variability.

3. Page 3, line 72: 20 years is indeed a substantial length, but climate variability at 20-year timescales still exist – it could be valuable to mention somewhere in the manuscript how the ADDA-v2 dataset can be supplemented to account for this in wind resource assessments.

There are indeed variabilities that exist within more climatological timescales, modulated by large-scale oscillations. Our data is not long enough to encompass all such variabilities. However, ADDA-v2 can be supplemented by other, longer datasets, such as CONUS404 (Rasmussen et al., 2024), which covers a period of 41 years between 1980-2020 to provide a more comprehensive means for examining variability on longer timescales. We will add this point to *Discussion*.

4. Page 4, line 98: Could you provide motivation for the choice of two months of spin-up? It is rather limited for the soil component as says a publication of Jerez et al (2020) [https://doi.org/10.1029/2019MS001945], definitely if simulations start in the cold months.

We understand the reviewer's concern and have read the paper by Jerez et al. (2020). Please note that this paper studied the topic of spin-up time when the RCM is driven by GCMs. In our practice, when running RCMs driven by GCMs, we use at least one-year spin-up time and run the model continuously without re-initialization. However, the simulation we presented in this manuscript is driven by reanalysis ERA5. We have evaluated ERA5's soil moisture by comparing against satellite observations and have found that the soil moisture in ERA5 is realistic (see figure below). This indicates that the soil moisture is not of concern in reanalysis driven runs as it would be in GCM driven runs. We have also evaluated the ADDA's soil moisture in this these simulations and they look reasonable as well. Additionally, there are many previous studies (e.g., Qian et al., 2003; Lucas-Picher et al., 2013; Pan et al., 1999) looking at model spin-up times, their impacts on model performance, and their effects when compared with using spectral nudging. In short, the findings show that reinitialization can reduce the model bias in the long term, which can achieve similar performance to spectral nudging. Therefore, in this study, due to the additional computational cost, we have not applied the nudging technique but instead use reinitialization to reduce potential bias. We will add clarification during revision.

References:

Qian, S. S., King, R. S. & Richardson, C. J. Two statistical methods for the detection of environmental thresholds. *Ecol. Model.* **166**, 87–97. https://doi.org/10.1016/S0304-3800(03)00097-8 (2003).

Lucas-Picher, P., Boberg, F., Christensen, J. H. & Berg, P. Dynamical downscaling with reinitializations: A method to generate fine scale climate datasets suitable for impact studies. *J. Hydrometeorol.* **14**, 1159–1174. https://doi.org/10.1175/JHM-D-12-063.1 (2013).

Pan, Z., Takle, E., Gutowski, W. & Turner, R. Long simulation of regional climate as a sequence of short segments. *Mon. Weather Rev.* **127**, 308–321. https://doi.org/10.1175/1520-0493(1999)127%3C0308:LSORCA%3E2.0.CO;2 (1999)

[Figure]

5. Page 4, line 122: On such a large domain, why is no spectral nudging employed for small wavenumbers? Has it been validated at some point that the properties of synoptic systems are indeed adequately reproduced? It would be good to comment on this.

Thanks for the comment. During the early stages of testing this model setup, we

had tested the effect of spectral nudging using varying wavenumbers, heights at and above which to use nudging, and nudging strength (as documented in Wang and Kotamarthi (2013)). While nudging is very useful when the spatial resolutions are very different between forcing data and RCM (e.g. when we use NCEP-R2 at 250km), the impacts of nudging for ERA5 for our 4km run were not as significant because the ERA5 is already at a very decent resolution (~30km) compared with other global reanalysis. Given the additional cost of employing spectral nudging, we opted not to use nudging for this simulation. In terms of the model performance of properties of synoptic systems, you can refer to the wind roses in Section 3.1.3. You can also see our precipitation performance and evaluation in Akinsonola et al. (2024) which also demonstrated that our 4km run captures the synoptic patterns well. We will add clarification to our revision.

References:

Wang, J., and V. R. Kotamarthi, 2013: Assessment of Dynamical Downscaling in Near-Surface Fields with Different Spectral Nudging Approaches Using the Nested Regional Climate Model (NRCM), *Journal of Applied Meteorology and Climatology,* 52, 1576–1591

Akinsanola, A. A., Jung, C., Wang, J., & Kotamarthi, V. R. (2024). Evaluation of precipitation across the contiguous United States, Alaska, and Puerto Rico in multi-decadal convection-permitting simulations. Scientific Reports, 14(1), 1238.

6. Page 5, table 1: were there any model options activated to account for subgrid-scale orography (e.g. topo_wind for YSU PBL scheme or the GWDO scheme) – would be good to mention this.

Yes, these were activated for these simulations, and our initial tests show that it can improve wind performance over complex terrain. We will mention this during revision.

7. Page 5: table 1: can you motivate the choice for 49 vertical levels? Low-level winds are usually quite sensitive to this choice. Perhaps sensitivity tests were conducted?

We tested the sensitivity of different vertical level configurations, mostly comparing to the choice for our previous, 12km simulations. We found that 50 levels perform better than the 38 levels we used previously for the 12km resolution simulations. In this simulation we also added many more layers below 1km. We have 18 σ levels below 1 km (8, 25, 42, 58, 75, 104, 147, 189, 231, 274, 317, 360, 403, 468, 555, 643, 777, and 957m above ground level). Depending on the needs and the use of this dataset, such as boundary layer physics, urban meteorology, we believe that we may be beneficial further with finer vertical resolution at the lower level. We will add some discussion during revision.

8. Page 5, line 138: are the observations also corrected for mast flow distortions? Perhaps good to comment on this and the implication on observational uncertainty.

For most of the observations which had only a single anemometer reading, no correction for mast flow distortion was performed. The orientation of the anemometers with respect to the towers was not commonly provided in the

metadata, and we did not wish to make corrective assumptions. For the occasional site where there were multiple anemometer readings at the same height, the maximum of the two wind speeds at each timestamp was selected to mitigate against mast flow distortion. We will add this information in the methodology (Section 2.2)

9. Page 11, line 267: "As discussed in (Section 3.1)" should be corrected.

Will correct during revision.

10. Page 11, line 277: I would not use "improvement" here.

We agree and will rephrase.

11. Page 13, line 293: So is this the r between the seasonally-averaged values? Or is it the seasonally-averaged value of daily r values? Sometimes in the manuscript this is not very clear.

The Pearson correlation coefficient in this section is for seasonally averaged diurnal cycles, rather than seasonally averaged r values for daily averages. We will clarify this.

12. Page 19, section 3.1.3 (wind roses): I agree that a good approximation of the wind rose is a first indication that synoptic winds are well captured. However, the manuscript would benefit from any additional analysis (or references to related studies) which looks at this in more detail. I mainly say this because the domain is very large and no nudging is used. If this cannot be provided, it would be good to mention that this has not been inspected in full detail.

We agree that the seasonal averaged wind-roses do provide model performance at the synoptic scale. Additionally, we argue that the evaluation at hub heights for specific locations provides more information than just at the synoptic scale. It provides more detailed information about the model's ability to capture the finer-scale wind patterns closer to the surface (at much lower heights than 850hPa/500hPa, where synoptic scale mechanisms are more present). During the revision, we plan to look at the diurnal cycles of wind direction in each season to add another component to a synoptic perspective. However, we are also open to suggestions from the reviewer for ideas for additional analysis.

13. Page 19, line 377: section index 3.1.3 is repeated here.. should be 3.1.4 I think?

Correct. Will fix it during revision.

14. Page 21, line 409: your prior analysis of seasonally-average diurnal cycles shows that statistically (not for specific days) the diurnal cycle is captured well. Here you seem to suggest to the reader not to use the sub-daily information of ADDA-v2. Is this not too critical? If you would feed the diurnal cycles of 500 winter days from ADDA-v2 to a wake model to check performance of a wind farm, I don't feel like

the output would not be trustworthy. So maybe rephrase this a bit or provide additional clarification.

We appreciate this comment and will clarify the context of this statement in the manuscript. We wanted to convey that ADDA-v2 is a climatological dataset and is not intended for weather-scale (e.g., day-to-day) evaluations. But for studies like the reviewer suggested, we believe the data and this analysis is still valuable.

15. Page 22, figure 6: As surface wind speeds are usually well below 10 ms-1 on average, RMSE does not give a good idea of how good the agreement actually is. Could you use the rRMSE here?

We agree and will adjust Figure 6 to display rRMSEs instead.

16. Page 22, figure 6: In the introduction you mention that wind datasets are important for risk assessments of high winds. Yet, for this surface wind speed evaluation, you focus on seasonal RMSE's. Why not also look at extremes? Is there perhaps a reason why this dataset is not intended for looking at extreme winds? Logically, for wind energy purposes one would be interested in winds higher up, so I'm curious what the surface wind speed output is for. You could add an analysis, or clarify why the presented analysis is sufficient.

Thanks for the comment and suggestion. Indeed, one of the motivations for developing this high-resolution data was for risk assessments associated with extreme weather events. We will include an extremes analysis during revision. For example, we can compare the hourly output of ADDA-v2 alongside the hourly ASOS data focusing on the 95th percentile of wind speeds to validate ADDA-v2's ability to capture extreme wind events. We are also open to any suggestions from the reviewer in terms of additional extreme analyses to the manuscript.

Please also note that we have a separate study that conducted assessments on wind extremes using ADDA-v2 focusing on tropical cyclones (TC) in the Atlantic basin. Generally, we find that the ADDA-v2 data can accurately capture the TC characteristics, including categories, intensities, frequencies and duration. This work is currently under review. We agree that, by including an extremes validation for inland CONUS, we can further demonstrate ADDA-v2's utility in wind-related risk assessments.

17. Page 24, figure 7: These indicators '7-2', '5-1' – where do these come from? These should be explained somewhere.

Apologies for the confusion. We will clarify what the indicators such as '7-1' or '5-2' mean. They refer to the different options for the dynamic vegetation and surface drag parameterizations that are provided within the Noah Multi-Parameterization land surface scheme.

18. Page 25, line 498: Instead of saying that high friction velocities correspond to weaker winds, could you explain a bit better to the reader why this is the case?

While friction velocity is not a scale for wind speed itself, they tend to have strong correlations. Friction velocity quantifies the turbulent momentum flux at the surface. Therefore, higher $u^*$ values correspond to more of the momentum being lost to the surface, leading to weaker wind speeds closer to the ground, especially in areas with high surface roughness. We can clarify this so that the connection between friction velocity and wind speeds is clear.

19. Page 26, section 3.4: An interesting addition. However, I am very interested to see the importance of the two components of model uncertainty: initialization and physics choices. I would expect that the lateral forcing would lead the initialization to not matter too much on annual statistics compared to physics parametrization choices. Please add this.

The hypothesis from the reviewer is correct. Over most locations, the choice of physics parameterizations shows a larger range of model outcomes when compared to the varying initialization conditions. We will be able to add a figure to (likely) a supplementary file to show the difference in magnitudes for internal variability (varying initialization times) and structure uncertainty (varying model physics parameterizations). Some figures highlighting this can be seen below, in which the standard deviation between the structure uncertainty ensembles is larger than that of the internal variability ensembles:

[Figure]

Figure R1. January diurnal cycle of 100m wind speed over four representative 6x6

grid regions using 10 ensemble members with varying initial conditions (dates and hours). The spread of these lines indicates the internal variability of our model over different regions.

[Figure]

Figure R2. Same as Figure R1, but the spread indicates structure uncertainty due to physics parameterizations, specifically looking at different land surface model and Planetary boundary layer schemes.

20. Page 28, figure 10: Is the inter-annual variability over a specific 2-week period something that wind farm operators are concerned about? As expected, this variability is very large. I think that the inter-annual variability on seasonal timescales might be good to add as well: e.g. how good or how bad a winter period can be in terms of wind resource over the lifetime of a wind farm. You could motivate your choice or add also the seasonal timescale to this analysis.

Thanks for the comment. We'd like to clarify that the purpose of presenting biweekly model variability is mostly driven by the motivation of showing model uncertainty. That is, with longer time scales, the model uncertainty will decrease. We have calculated model uncertainty for weekly, biweekly and monthly timescales, and we chose to show bi-weekly here. The model uncertainty at the weekly scale is even larger than bi-weekly scale; contrarily, model uncertainty at monthly and seasonal scales are much smaller than the bi-weekly scale. We agree with the reviewer that the magnitude of interannual variability for the bi-weekly timescale might be less useful than seasonal timescales for wind resource evaluations. So, we will include interannual variability at seasonal scale likely in a supplementary file during revision.

21. Page 29, line 553: Is the lower inter-annual variability in summer not simply a consequence of lower wind speeds? Relatively speaking, the variability could be as large as for winter or even larger.

We agree with the reviewer on both points they brought up. We can plot maps to show the relative values of these interannual variability (to the actual wind speeds)

22. Page 30, line 601: I get a "404 not found" when pasting the link to reach the hub height wind data. Please make sure that a reliable pointer is available to access the data and that some documentation is available there. Perhaps also for the full ADDA-v2 data: include an e-mail / link where to request access.

That is very strange. We double checked and the links were accessible and took the user to the intended destination. The following websites are where the links should take you:

WTK-LED Climate API | NREL: Developer Network
WDH: Wind Data Hub

We will ensure that the links are functional during revision. We will also include the information necessary to request access to the full ADDA-v2 dataset.

**General feedback**

Overall, this manuscript is well written, well-structured and appears carefully worked through with nice looking figures. The work describes a new 4km, 20y mesoscale dataset covering North America with extensive validation using met towers and surface stations and ensemble analysis for a selected period.

Downscaling of global reanalysis models using mesoscale models like WRF is well covered in the literature as well as the improvements it provides relative to the global models. Hence, the novelty of the approach in this manuscript may be disputed given that it has poorer resolution compared to the previous work of Draxl et al. (2015).
However, the open access to the large dataset and the extensive validation effort including ensemble analysis justifies the publication.

We appreciate all the insightful comments which we believe will further improve our manuscript. Please find our response to each individual reviewer's comment in the following section.

**General comments**

In general, I would like to question if the selected validation metrics for wind speed (r, RMSE, rRMSE, OVL) provide sufficient complementary insight. In my view, these metrics overlap too much in what they measure and none of them allow for distinction between systematic errors (biases) and fluctuating errors. I suggest including a simple metric like mean (bias) error to cover this important aspect and re-reconsider if each of the other metrics contribute enough additional insight to remain in the paper. A metric should be included only if characteristic error structures can be inferred from it – to move beyond being merely descriptive.

Thanks for the comment. We agree that mean bias has been used very commonly in wind data evaluation studies, and it is very effective if there is a systematic model bias. For example, in our model configuration, we found that there is a systematic high bias when using Noah Land surface model (compared with NoahMP) over the Midwest region. In this case, using this error metric can effectively show systematic bias. However, in the case of locations or regions that do not exhibit such systematic bias, and the model bias varies with time - for example, one year shows negative bias, and another year shows positive bias - our concern was that the mean bias may be smoothed out and may show a misleading conclusion that the model performs well. We will investigate this in depth and compare mean bias with RMSEs and see whether they have similar conclusions over different regions of our model domain. In fact, our collaborative study by Sheridan et al. 2024 has looked at the ADDA-v2 data and many other datasets over coastal regions of US, employing mean bias as an error metric. This study will help with the investigation as well. We will add discussion with any interesting findings.

Regarding the metrics employed in this study, each was chosen to offer a unique component to the validation. Initially, looking at the full distribution of wind speeds, the PDFs paired with these overlap ratios (OVLs) were used to demonstrate the degree of similarity between model and observational wind speeds without considering the time dimension. These PDFs can visually convey any systematic biases present within the model.

Next, RMSEs and rRMSEs were then paired with the diurnal cycle plots, which now consider the time dimension, unlike the PDFs. We chose RMSEs to test how close ADDA-v2's wind speeds were to observations in the absolute sense and include rRMSE to show the magnitude of error relative to the wind speeds themselves. In addition, while RMSEs could demonstrate that the model performs very well in terms of magnitude, it is not able to show whether the model captures the correct timing of the wind speed minimums and maximums. Thus, we also use Pearson's correlation coefficient. Further, we use wind-roses to examine the wind speed and corresponding wind direction to ensure the model captures the physics and the seasonality correctly.

I suggest reducing the mostly summarising parts (section 3) with long descriptions and lists of numbers in the text. Please also consider additional summary table(s) for better overview and readability.

Thanks for the comments. Following your suggestion, we will try to condense the result section during revision. Regarding the summary table, all error metrics that were calculated and mentioned from Section 3.1.1 to Section 3.1.3 are summarized in Table 3. If the reviewer has any other suggestions about how to revise this table, we would be happy to address.

The paper should include consideration/discussion of the effect of not accounting microscale effects. A 4km model effectively resolves scales from 20-30km and up. How is this expected to affect presented results, when validating the model against measurements that include significant effects on finer scales, which may be very strong at 10m agl.?

Thank you for the insight. We agree that, although the model is at a grid spacing of 4km, it is not able to resolve the energy at scales finer than 10-20km, as discussed in Müller et al. (2024, Figure 11) Skamarock (2004) and Larsén et al. (2012). This means that our model can capture wind variability at 10-20km scale but cannot resolve the wind variability at 4-10km scale that exists within the observational data, especially at the near-surface level. Thus, the bias we see in our analysis is not only due to model physics or configuration, but also the fact that the model at this spatial resolution is not able to resolve the variability at such fine scales. We will add a discussion acknowledging such limitations in our model evaluations.

Reference:
Skamarock, W. C.: Evaluating mesoscale NWP models using kinetic energy spectra, Mon. Weather Rev., 132, 3019–3032, https://doi.org/10.1175/MWR2830.1, 2004

Larsén, X. G., Ott, S., Badger, J., Hahmann, A. N., and Mann, J.: Recipes for correcting the impact of effective mesoscale resolution on the estimation of extreme winds, J. Appl. Meteorol. Clim., 51, 521–533, https://doi.org/10.1175/JAMC-D-11-090.1, 2012.

Müller, S., Larsén, X. G., and Verelst, D. R.: Tropical cyclone low-level wind speed, shear, and veer: sensitivity to the boundary layer parametrization in the Weather Research and Forecasting model, Wind Energy. Sci., 9, 1153–1171, https://doi.org/10.5194/wes-9-1153-2024, 2024.

Argumentation that the selected ensemble runs represent model uncertainty should be strengthened, this currently is an implied assumption. Does the spread across the selected and boot-strapped ensambles really represent actual model uncertainty?

Thanks for the question. We agree that it is very challenging for numerical simulations at such a high resolution over a large domain to capture *all* model uncertainty. So, we aimed to design the presented model configurations to represent a robust sample of model uncertainty. We chose to perturb the Planetary Boundary Layer Scheme and the land surface model for the "structure uncertainty" simulations because they have the most significant influence on generating variability within near-surface winds (Draxl et al., 2014; Yang et al., 2017). Of course, we understand that many other physics parameterizations can cause different model solutions as well. For internal variability, we conducted the minimum number required for quantifying the uncertainty (Wang et al. 2017).

We can make this clearer in the manuscript by providing a brief discussion justifying the selected model configurations used for this model sensitivity analysis. We can also note that recent advances in machine-learning (ML) based surrogate model or numerical+ML hybrid modeling may provide a more comprehensive means of quantifying model uncertainty (Tunnell et al, 2023; Di Santo et al., 2025) given the much faster calculation they can do.

Referring to the response to Reviewer #1, our data, alongside other existing datasets with more extensive time periods (albeit more limited domains) can also provide a more comprehensive understanding of model uncertainty and variability.

References:

Tunnell, M., Bowman, N., & Carrier, E. (2023). Fast Gaussian process emulation of Mars Global Climate Model. *Earth and Space* Science, 10, e2022EA002743, https://doi.org/10.1029/2022EA002743

Di Santo, D., He, C., Chen, F., & Giovannini, L. (2025). ML-AMPSIT: Machine Learning-based Automated Multi-method Parameter Sensitivity and Importance analysis Tool. *Geoscientific Model Development, 18*, 433–459. https://doi.org/10.5194/gmd-18-433-2025

The limitations and uncertainty of the observations used in the validation should be discussed either in section 2.2 or section 4.

We agree and will add a discussion to address this during the revision process. Such limitation and uncertainty include representativeness errors, in which there could be a scale mismatch between the hyperlocal measurement conditions of the anemometer and the broader model grid cell, environmental effects such as land use, obstructions, or elevation effects, or the temporal sampling methods of the observational data and the inherent uncertainties associated with that.

Reviewer 1 additionally inquired about whether the observations were corrected for mast flow distortions. Much of the hub-height observational data we worked with did not have the orientation of the anemometers with respect to surrounding structures/towers. Therefore, we did not want to make corrective assumptions and potentially incite additional biases into the observational data. However, we do agree that it is important to acknowledge the limitations of the observational data itself.

**Some detailed comments**

Page 2, line 63:    It should be mentioned here that ERA5 is initial/boundary model in addition to the info in table 1, on page 5.

Will include this during revision.

Page 10, line 223: Explain "internal variability" and "structure uncertainty" in more detail, and why 10 and 6 ensemble members, respectively, was decided upon.

We can elaborate on the specific definitions of internal variability and structure uncertainty, as well as why both were included in our analysis of model uncertainty. Ensemble members testing for internal variability had varying initialization times, but identical physics parameterizations, while ensemble members testing for structure uncertainty had the same initialization conditions, but varying model physics. We will discuss this in greater detail in Section 2.1. Referencing the response to general comment 4, we will also expand upon the specific justifications of the selected "structure uncertainty" ensemble members.

Page 14-15, fig. 3:    A legend is missing for the plots.

Will add during revision.

Page 15, line 340:  Interpolation in wind direction simply requires conversion of wind direction to components which may be interpolated similar to the wind speeds, and then converted back to wind directions.

Thanks for the comment. There have been studies demonstrating that the interpolation methods used for windspeed/direction profiles face challenges in regions of heterogeneous terrain and rough surfaces (e.g., Lalic et al., 2012). We will test this interpolation for simple terrains and then for more complex terrains, in which wind direction profiles are typically more complicated. We will add discussions or analysis with any interesting findings.

Reference
Lalic, Branislava & Mihailovic, Dragutin & Kapor, Darko. (2012). Limitations and Uncertainties in the Logarithmic Wind Profile Above Very Rough Surfaces.

Page 25, line 489:    Friction velocity is denoted using $u_*$ and not $u^*$.

Will fix.

Page 26, line 503:    Explain why "high friction velocities correspond to weaker winds"

We can clarify this so that the connection between friction velocity and wind speeds is clear. While friction velocity is not a direct scale for wind speed itself, they tend to have strong correlations. Friction velocity quantifies the turbulent momentum flux at the surface. Therefore, higher $u_*$ values correspond to more of the momentum being lost to the surface, leading to weaker wind speeds closer to the ground, especially in areas with high surface roughness.

---

## Author Response (AR1)

**Overall feedback**

This manuscript provides an extensive evaluation of the ADDA-v2 climatological wind dataset for North America. The manuscript text is logically structured and well-written and the figures are clear. A large set of measurement stations and different aspects of the model performance are considered over a range of timescales. The analysis goes into sufficient depth and an uncertainty analysis is included. It is also valuable that a comparison is done to the widely used dataset used to drive the model runs, i.e. ERA5 to see where the downscaled dataset significantly improves on the driving data. However, some choices or sections in the manuscript would benefit from changes or additions, which I will detail below.

Thank you for the insightful comments which we find valuable in improving our manuscript. We have responded to each of the following general and specific comments (in blue font) and have made appropriate adjustments during the revision process.

**General comments**

1. It would be valuable if the findings of this evaluation (and the ADDA-v2 dataset itself) would be compared to other evaluation studies (and other datasets), preferably for the region. See also specific comment nr. 2. Hence, some references should be added in the discussion section (or somewhere else) to better situate this wind product in relation to other datasets and also how the evaluation/performance differs from other studies.

We agree and have contextualized the added value of ADDA-v2 to existing datasets that cover a similar region, including the wind dataset discussed by Draxl et al. (2015a, b). Such studies were already mentioned in the Introduction (Draxl et al., 2015a, b; Gensini et al., 2023; Liu et al., 2017; Rasmussen et al., 2024) but we have readdressed it in Section 4 *Discussion* to situate ADDA-v2 in relation to these other high-resolution datasets. We've also added a discussion mentioning complementary studies of ADDA-v2 and its performance in other wind-related contexts, including coastal locations and offshore for tropical cyclones.

Passage taken from manuscript that demonstrates these discussions:

"Additionally, while this validation focused more on inland regions, Sheridan et al. (2025) has evaluated ADDA-v2's performance over coastal locations. Tobias-Tarsh et al. (2025) has evaluated ADDA-v2's performance in wind-related extremes in the context of tropical cyclones over the North Atlantic Basin.

Other studies exist that introduce wind datasets and validate them against observations. For instance, Draxl et al. (2015) documented a 7-year wind dataset with a grid spacing of 2km, primarily focused on wind power evaluations over CONUS and included a limited meteorological validation using 6 tall masts and 3 buoys. Rasmussen et al. (2024) performed validations on a 42-year period, 4km dataset on its near-surface (10m) wind speeds with underestimation especially over complex terrain. While these datasets provide their own unique utility, ADDA-v2 offers a powerful combination of a reasonably long time period with a large spatial domain. By comprehensively validating ADDA-v2's wind speeds and directions using an extensive network of near-surface observations and a diverse set of hub-height observations, this evaluation can provide insight for both

climatological studies and wind resource assessments. Yet, all these datasets can be used collectively, complementing one another with their unique characteristics and allowing for a more comprehensive view of model uncertainty and longer-term variability."

**Specific comments**

1. Page 2, line 40: I recommend adding that fine-scale surface variations (topography, land cover, …) are also better represented.

   We agree and have added this.

2. Page 2, Line 60: Considering that this wind dataset of Draxl et al. (2015a,b) exists and has an even higher spatial resolution, could you add somewhere in the manuscript why this new data product ADDA-v2 is perhaps a better choice going forward?

   Referring to the response to the general comment, we have included a component to the discussion that addresses existing datasets and outlines the added merit of ADDA-v2 in climatological wind studies or wind-resource assessments. We plan to highlight the more extensive temporal coverage of ADDA-v2 as well as the unique spatial domain that spans into regions typically outside the domain of other datasets, such as Alaska, Mexico, and Caribbean islands. Additionally, the ensemble analysis is useful for model uncertainty quantification; and the extensive temporal coverage can help examine inter-annual variability.

3. Page 3, line 72: 20 years is indeed a substantial length, but climate variability at 20-year timescales still exist – it could be valuable to mention somewhere in the manuscript how the ADDA-v2 dataset can be supplemented to account for this in wind resource assessments.

   There are indeed variabilities that exist within more climatological timescales, modulated by large-scale oscillations. Our data is not long enough to encompass all such variabilities. However, ADDA-v2 can be supplemented by other, longer datasets, such as CONUS404 (Rasmussen et al., 2024), which covers a period of 41 years between 1980-2020 to provide a more comprehensive means for examining variability on longer timescales. We will add this point to *Discussion*.

4. Page 4, line 98: Could you provide motivation for the choice of two months of spin-up? It is rather limited for the soil component as says a publication of Jerez et al (2020) [https://doi.org/10.1029/2019MS001945], definitely if simulations start in the cold months.

   We understand the reviewer's concern and have read the paper by Jerez et al. (2020). Please note that this paper studied the topic of spin-up time when the RCM is driven by GCMs. In our practice, when running RCMs driven by GCMs, we use at least one-year spin-up time and run the model continuously without re-initialization. However, the simulation we presented in this manuscript is driven by reanalysis ERA5. We have evaluated ERA5's soil moisture by comparing against satellite observations and have

found that the soil moisture in ERA5 is realistic (see figure below). This indicates that the soil moisture is not of concern in reanalysis driven runs as it would be in GCM driven runs. We have also evaluated the ADDA's soil moisture in this these simulations and they look reasonable as well. Additionally, there are many previous studies (e.g., Qian et al., 2003; Lucas-Picher et al., 2013; Pan et al., 1999) looking at model spin-up times, their impacts on model performance, and their effects when compared with using spectral nudging. In short, the findings show that reinitialization can reduce the model bias in the long term, which can achieve similar performance to spectral nudging. Therefore, in this study, due to the additional computational cost, we have not applied the nudging technique but instead use reinitialization to reduce potential bias. We will add clarification during revision.

[Figure]

Figure R1. Seasonal averages of soil moisture across CONUS for ADDA-v2 (2nd column) and ERA5 (3rd column) compared against gridded observational data (1st column) for the year 2005.

References:

Qian, S. S., King, R. S. & Richardson, C. J. Two statistical methods for the detection of environmental thresholds. *Ecol. Model.* **166**, 87–97. https://doi.org/10.1016/S0304-3800(03)00097-8 (2003).

Lucas-Picher, P., Boberg, F., Christensen, J. H. & Berg, P. Dynamical downscaling with reinitializations: A method to generate fine scale climate datasets suitable for impact studies. *J. Hydrometeorol.* **14**, 1159–1174. https://doi.org/10.1175/JHM-D-12-063.1 (2013).

Pan, Z., Takle, E., Gutowski, W. & Turner, R. Long simulation of regional climate as a sequence of short segments. *Mon. Weather Rev.* **127**, 308–321. https://doi.org/10.1175/1520-0493(1999)127%3C0308:LSORCA%3E2.0.CO;2 (1999)

5. Page 4, line 122: On such a large domain, why is no spectral nudging employed for small wavenumbers? Has it been validated at some point that the properties of synoptic systems are indeed adequately reproduced? It would be good to comment on this.

Thanks for the comment. During the early stages of testing this model setup, we had tested the effect of spectral nudging using varying wavenumbers, heights at and above which to use nudging, and nudging strength (as documented in Wang and Kotamarthi (2013)). While nudging is very useful when the spatial resolutions are very different between forcing data and RCM (e.g. when we use NCEP-R2 at 250km), the impacts of nudging for ERA5 for our 4km run were not as significant because the ERA5 is already at a very decent resolution (~30km) compared with other global reanalysis. Given the additional cost of employing spectral nudging, we opted not to use nudging for this simulation. In terms of the model performance of properties of synoptic systems, you can refer to the wind roses in Section 3.1.3. You can also see our precipitation performance and evaluation in Akinsonola et al. (2024) which also demonstrated that our 4km run captures the synoptic patterns well. We will add clarification to our revision.

References:

Wang, J., and V. R. Kotamarthi, 2013: Assessment of Dynamical Downscaling in Near-Surface Fields with Different Spectral Nudging Approaches Using the Nested Regional Climate Model (NRCM), *Journal of Applied Meteorology and Climatology,* 52, 1576–1591

Akinsanola, A. A., Jung, C., Wang, J., & Kotamarthi, V. R. (2024). Evaluation of precipitation across the contiguous United States, Alaska, and Puerto Rico in multi-decadal convection-permitting simulations. Scientific Reports, 14(1), 1238.

6. Page 5, table 1: were there any model options activated to account for subgrid-scale orography (e.g. topo_wind for YSU PBL scheme or the GWDO scheme) – would be good to mention this.

Yes, these were activated for these simulations, and our initial tests show that it can improve wind performance over complex terrain. We have added this information to the method section.

7. Page 5: table 1: can you motivate the choice for 49 vertical levels? Low-level winds are usually quite sensitive to this choice. Perhaps sensitivity tests were conducted?

We tested the sensitivity of different vertical level configurations, mostly comparing to the choice for our previous setup for 12km simulations. We found that 50 levels perform better than the 38 levels we used previously. In this simulation we also added many more layers below 1km. We have 18 σ levels below 1 km (8, 25, 42, 58, 75, 104, 147, 189, 231, 274, 317, 360, 403, 468, 555, 643, 777, and 957m above ground level) to make sure the hub-height winds are calculated instead of extrapolated. Depending on the needs and the use of this dataset, such as boundary layer physics, urban meteorology, it may be beneficial

to have this higher vertical resolution at the lower level. We have added this information in the methodology (Section 2.2).

8. Page 5, line 138: are the observations also corrected for mast flow distortions? Perhaps good to comment on this and the implication on observational uncertainty.

    For most of the observations which had only a single anemometer reading, no correction for mast flow distortion was performed. The orientation of the anemometers with respect to the towers was not commonly provided in the metadata, and we did not wish to make corrective assumptions. For the occasional site where there were multiple anemometer readings at the same height, the maximum of the two wind speeds at each timestamp was selected to mitigate against mast flow distortion. We've added this information in the methodology (Section 2.2)

9. Page 11, line 267: "As discussed in (Section 3.1)" should be corrected.

    Corrected

10. Page 11, line 277: I would not use "improvement" here.

    Rephrased

11. Page 13, line 293: So is this the r between the seasonally-averaged values? Or is it the seasonally-averaged value of daily r values? Sometimes in the manuscript this is not very clear.

    The Pearson correlation coefficient in this section is for seasonally averaged diurnal cycles, rather than seasonally averaged r values for daily averages. We have included clarification at the start of Section 3.1.2

12. Page 19, section 3.1.3 (wind roses): I agree that a good approximation of the wind rose is a first indication that synoptic winds are well captured. However, the manuscript would benefit from any additional analysis (or references to related studies) which looks at this in more detail. I mainly say this because the domain is very large and no nudging is used. If this cannot be provided, it would be good to mention that this has not been inspected in full detail.

    We agree that the seasonal averaged wind-roses do provide an indication of model performance at the synoptic scale. Additionally, we argue that the evaluation at hub heights for specific locations provides more information than just at the synoptic scale. It provides more detailed information about the model's ability to capture the finer-scale wind patterns closer to the surface (at much lower heights than 850hPa/500hPa (where synoptic scale mechanisms are more present). During the revision, we have also plotted diurnal cycles of wind direction in each season to add another component to a synoptic perspective. However, we are also open to suggestions from the reviewer for ideas for additional analysis.

    You can find our diurnal wind direction subplots that correspond to each of the wind rose

subplots in the Supplemental document.

13. Page 19, line 377: section index 3.1.3 is repeated here.. should be 3.1.4 I think?

    Corrected

14. Page 21, line 409: your prior analysis of seasonally-average diurnal cycles shows that statistically (not for specific days) the diurnal cycle is captured well. Here you seem to suggest to the reader not to use the sub-daily information of ADDA-v2. Is this not too critical? If you would feed the diurnal cycles of 500 winter days from ADDA-v2 to a wake model to check performance of a wind farm, I don't feel like the output would not be trustworthy. So maybe rephrase this a bit or provide additional clarification.

    We appreciate this comment. We wanted to convey that ADDA-v2 is a climatological dataset and is not intended for weather-scale (e.g., day-to-day) evaluations. But for studies like the reviewer suggested, we believe the data and this analysis is still valuable. We have added a couple sentences to the beginning of Section 3.1.5 to express this.

15. Page  22, figure 6: As surface wind speeds are usually well below 10 ms-1 on average, RMSE does not give a good idea of how good the agreement actually is. Could you use the rRMSE here?

    We agree and have adjusted the figure to show relative mean biases instead.

16. Page 22, figure 6: In the introduction you mention that wind datasets are important for risk assessments of high winds. Yet, for this surface wind speed evaluation, you focus on seasonal RMSE's. Why not also look at extremes? Is there perhaps a reason why this dataset is not intended for looking at extreme winds? Logically, for wind energy purposes one would be interested in winds higher up, so I'm curious what the surface wind speed output is for. You could add an analysis, or clarify why the presented analysis is  sufficient.

    Thanks for the comment and suggestion. Indeed, one of the motivations for developing this high-resolution data was for risk assessments associated with extreme weather events. We have a separate study that conducted assessments on wind extremes using ADDA-v2 focusing on tropical cyclones (TC) in the Northern Atlantic basin (Tobias-Tarsh et al. 2025). Generally, we find that the ADDA-v2 data can accurately capture the TC characteristics, including categories, intensities, frequencies and duration, and can do a much more reasonable job than ERA5 especially when using wind speed to define the hurricane category. When using sea level pressure, ERA5 performs reasonably well too. We have added this information in the Discussion mentioning this research. We also plan to perform more general wind-related extremes assessments over land in future studies. This is also mentioned in Section 4 *Discussion*

    References

    Tobias-Tarsh, L., Jung, C., Wang, J., Bobde, V., Akinsanola, A. A., and Kotamarthi, V. R.: Evaluation of North Atlantic Tropical Cyclones in a Convection-Permitting Regional Climate

Simulation, EGUsphere [preprint], https://doi.org/10.5194/egusphere-2025-1805, 2025.

17. Page 24, figure 7: These indicators '7-2', '5-1' – where do these come from? These should be explained somewhere.

    Apologies for the confusion. They refer to the different options for the dynamic vegetation and surface drag parameterizations that are provided within the Noah Multi-Parameterization land surface scheme.

    We have added a sentence in Section 2.1 explaining that we've perturbed the options for dynamic vegetation and surface layer drag coefficient calculation within the Noah-MP LSM for the sensitivity experiments we conducted. Also, we added a note within the figure caption for clarification.

18. Page 25, line 498: Instead of saying that high friction velocities correspond to weaker winds, could you explain a bit better to the reader why this is the case?

    While friction velocity is not a scale for wind speed itself, they tend to have strong correlations. Friction velocity quantifies the turbulent momentum flux at the surface. Therefore, higher $u_*$ values correspond to more of the momentum being lost to the surface, leading to weaker wind speeds closer to the ground, especially in areas with high surface roughness. We've added clarification to the section discussing friction velocity so that the connection between friction velocity and wind speeds is clear.

19. Page 26, section 3.4: An interesting addition. However, I am very interested to see the importance of the two components of model uncertainty: initialization and physics choices. I would expect that the lateral forcing would lead the initialization to not matter too much on annual statistics compared to physics parametrization choices. Please add this.

    The hypothesis from the reviewer is correct. Over most locations, the choice of physics parameterization shows a larger range of model outcomes when compared to the varying initial times (when the model simulation was initiated). We have added a figure to the Supplementary file to show the difference in magnitudes for internal variability (varying initialization times) and structure uncertainty (varying model physics parameterizations). Some figures highlighting this can be seen below, in which the standard deviation between the structure uncertainty ensembles is larger than that of the internal variability ensembles:

[Figure]

Figure R2. January diurnal cycle of 100m wind speed over four representative 6x6 grid regions using 10 ensemble members with varying initial conditions (dates and hours). The spread of these lines indicates the internal variability of our model over different regions.

[Figure]

Figure R3. Same as Figure R2, but the spread indicates structure uncertainty due to physics parameterizations, specifically looking at different land surface model and Planetary boundary layer schemes.

20. Page 28, figure 10: Is the inter-annual variability over a specific 2-week period something that wind farm operators are concerned about? As expected, this variability is very large. I think that the inter-annual variability on seasonal timescales might be good to add as well: e.g. how good or how bad a winter period can be in terms of wind resource over the lifetime of a wind farm. You could motivate your choice or add also the seasonal timescale to this analysis.

Thanks for the comment. We'd like to clarify that the purpose of presenting biweekly model variability is mostly driven by the motivation of showing how model uncertainty changes with time scales. That is, with longer time scales, the model uncertainty will decrease. We have calculated model uncertainty for weekly, biweekly and monthly timescales, and we chose to show bi-weekly here. The model uncertainty at the weekly scale is even larger than bi-weekly scale; contrarily, model uncertainty at monthly and seasonal scales are much smaller than the bi-weekly scale. We agree with the reviewer that the magnitude of interannual variability for the bi-weekly timescale might be less useful than seasonal timescales for wind resource evaluations. So, we have included interannual variability at seasonal scale in the supplementary file during revision. Findings are also included in the result section (Section 3.4).

21. Page 29, line 553: Is the lower inter-annual variability in summer not simply a consequence of lower wind speeds? Relatively speaking, the variability could be as large as for winter or even larger.

We agree with the reviewer on both points they brought up. We have plotted maps to show the relative values of interannual variability (to the actual wind speeds)

22. Page 30, line 601: I get a "404 not found" when pasting the link to reach the hub height wind data. Please make sure that a reliable pointer is available to access the data and that some documentation is available there. Perhaps also for the full ADDA-v2 data: include an e-mail / link where to request access.

That is very strange. We double checked and the links were accessible and took the user to the intended destination. The following websites are where the links should take you:

WTK-LED Climate API | NREL: Developer Network
WDH: Wind Data Hub

We have ensured that the links are functional. We will also include the information necessary to request access to the full ADDA-v2 dataset.

**General feedback**

Overall, this manuscript is well written, well-structured and appears carefully worked through with nice looking figures. The work describes a new 4km, 20y mesoscale dataset covering North America with extensive validation using met towers and surface stations and ensemble analysis for a selected period.

Downscaling of global reanalysis models using mesoscale models like WRF is well covered in the literature as well as the improvements it provides relative to the global models. Hence, the novelty of the approach in this manuscript may be disputed given that it has poorer resolution compared to the previous work of Draxl et al. (2015).
However, the open access to the large dataset and the extensive validation effort including ensemble analysis justifies the publication.

We appreciate all the insightful comments which we believe have improved our manuscript. Please find our response to each individual reviewer's comment in the following section. We have also significantly reduced the text in the results yet maintained the highlights of discussion.

**General comments**

In general, I would like to question if the selected validation metrics for wind speed (r, RMSE, rRMSE, OVL) provide sufficient complementary insight. In my view, these metrics overlap too much in what they measure and none of them allow for distinction between systematic errors (biases) and fluctuating errors. I suggest including a simple metric like mean (bias) error to cover this important aspect and re-reconsider if each of the other metrics contribute enough additional insight to remain in the paper. A metric should be included only if characteristic error structures can be inferred from it – to move beyond being merely descriptive.

Thanks for the comment. We agree that mean bias has been used very commonly in wind data evaluation studies, and it is very effective if there is a systematic model bias. For example, in our model configuration, we found that there is a systematic high bias in near-surface wind when using Noah Land surface model (compared with NoahMP) over the Midwest region (as shown in Figure 8). In this case, using this error metric can effectively show systematic bias. However, in the case of locations or regions that do not exhibit such systematic bias, and the model bias varies with time - for example, one year shows negative bias, and another year shows positive bias - our concern was that the mean bias may be smoothed out and may show a misleading conclusion that the model performs well. While we had this concern, we investigated this in depth during revision and compared mean bias with our other metrics. We found the mean bias shows similar conclusion with PDFs about the systematic underestimation of ERA5. However, because mean bias was calculated with the time dimension, it does add value on top of PDFs to allow us to better understand model bias. There is not much systematic bias in ADDA-v2 but a slight overestimation over some sites. We have added a section and a figure with 3 panels about this analysis in Section 3.1.2.

Regarding the other metrics employed in this study, each was chosen to offer a unique component to the validation. Initially, looking at the full distribution of wind speeds, the PDFs

paired with these overlap ratios (OVLs) were used to demonstrate the degree of similarity between model and observational wind speeds without considering the time dimension. These PDFs can visually convey any systematic biases present within the model.

Next, RMSEs and rRMSEs were then paired with the diurnal cycle plots, which now consider the time dimension, unlike the PDFs. We chose RMSEs to test how close ADDA-v2's wind speeds were to observations in the absolute sense and include rRMSE to show the magnitude of error relative to the wind speeds themselves. In addition, while RMSEs could demonstrate that the model performs very well in terms of magnitude, it is not able to show whether the model captures the correct timing of the wind speed minimums and maximums. Thus, we also use Pearson's correlation coefficient. Further, we use wind-roses to examine the wind speed and corresponding wind direction to ensure the model captures the physics and the seasonality correctly.

I suggest reducing the mostly summarising parts (section 3) with long descriptions and lists of numbers in the text. Please also consider additional summary table(s) for better overview and readability.

Thanks for the comments. Following your suggestion, we have significantly reduced the text and description in the result section while still highlighting model performance in representative regions (e.g., flat, mountains, Alaska and Puerto Rico). While the text is condensed, statistics are still summarized in Table 3 for all metrics, providing detailed evaluations for both ADDA and ERA5 datasets.

The paper should include consideration/discussion of the effect of not accounting microscale effects. A 4km model effectively resolves scales from 20-30km and up. How is this expected to affect presented results, when validating the model against measurements that include significant effects on finer scales, which may be very strong at 10m agl.?

Thank you for the insight. We agree that, although the model uses a grid spacing of 4km, it cannot fully resolve the energy spectrum or variability at scales finer than 10-20km, as discussed in Müller et al. (2024, Figure 11), Skamarock (2004) and Larsén et al. (2012). This means that our model cannot capture the wind variability at 4-10km scale that exists within observations data, particularly at the near-surface level where variability tends to be larger. Capturing such variability in observations would require continuous gridded data, such as those from radar or satellites. However, the observational data we use are from individual point locations and do not represent spatial variability in the surrounding area. To make the evaluations more robust, we could expand the current approach to include multiple model grid cells surrounding each observation site, rather than using only the closest grid cell. This would allow us to characterize a range of modeled winds around the observation sites and better represent model spatial variability. An example of this approach is shown by Müller (2025) for typhoon evaluation using mesoscale models and lidar data. We have added a discussion of this in the revised manuscript.

[Figure]

Figure R3: Top: Simulated track of Typhoon Maysak (2020) according to the JMA best track dataset (black solid line) and as obtained by a mesoscale simulation (black dashed line). The red and blue triangles indicate the locations of the two lidars. Black dots surrounded the red and blue triangles are grid points from a mesoscale simulation. Bottom: time series of wind speeds from lidars, and model simulations. dark red line is from the closest grid point, while the gray lines are from the surrounded grid points which show a large variability in wind speeds. Source: Müller (2025)

Reference:
Skamarock, W. C.: Evaluating mesoscale NWP models using kinetic energy spectra, Mon. Weather Rev., 132, 3019–3032, https://doi.org/10.1175/MWR2830.1, 2004

Larsén, X. G., Ott, S., Badger, J., Hahmann, A. N., and Mann, J.: Recipes for correcting the impact of effective mesoscale resolution on the estimation of extreme winds, J. Appl. Meteorol. Clim., 51, 521–533, https://doi.org/10.1175/JAMC-D-11-090.1, 2012.

Müller, S., Larsén, X. G., and Verelst, D. R.: Tropical cyclone low-level wind speed, shear, and veer: sensitivity to the boundary layer parametrization in the Weather Research and Forecasting model, Wind Energy. Sci., 9, 1153–1171, https://doi.org/10.5194/wes-9-1153-2024, 2024.

Müller 2025. Typhoon wind and turbulence structure, and its impact on wind energy application. PhD thesis. Department of Wind and Energy Systems, DTU Wind.

Argumentation that the selected ensemble runs represent model uncertainty should be strengthened, this currently is an implied assumption. Does the spread across the

selected and boot-strapped ensambles really represent actual model uncertainty?

Thanks for the question. We agree that it is very challenging for numerical simulations at such a high resolution over a large domain to capture *all* model uncertainty. So, we aimed to design the presented model configurations to represent a robust sample of model uncertainty. We chose to perturb the Planetary Boundary Layer Scheme and the land surface model for the "structure uncertainty" simulations because they have the most significant influence on generating variability within near-surface winds (Draxl et al., 2014; Yang et al., 2017). Of course, we understand that many other physics parameterizations can cause different model solutions as well. For internal variability, we conducted the minimum number required for quantifying the uncertainty (Wang et al. 2017).

We have made this clearer in the manuscript by providing a brief discussion justifying the selected model configurations used for this model sensitivity analysis. We've also noted that recent advances in machine-learning (ML) based surrogate model or numerical + ML hybrid modeling may provide a more comprehensive means of quantifying model uncertainty (Tunnell et al, 2023; Di Santo et al., 2025) given the much faster calculation they can do.

Referring to the response to Reviewer #1, our data, alongside other existing datasets with more extensive time periods (albeit more limited domains) can also provide a more comprehensive understanding of model uncertainty and variability.

References:

Tunnell, M., Bowman, N., & Carrier, E. (2023). Fast Gaussian process emulation of Mars Global Climate Model. *Earth and Space* Science, 10, e2022EA002743, https://doi.org/10.1029/2022EA002743

Di Santo, D., He, C., Chen, F., & Giovannini, L. (2025). ML-AMPSIT: Machine Learning-based Automated Multi-method Parameter Sensitivity and Importance analysis Tool. *Geoscientific Model Development, 18*, 433–459. https://doi.org/10.5194/gmd-18-433-2025

The limitations and uncertainty of the observations used in the validation should be discussed either in section 2.2 or section 4.

We agree and have added a discussion to address this in Section 2.2 (Observational datasets). Such limitation and uncertainty include representativeness errors, in which there could be a scale mismatch between the hyperlocal measurement conditions of the anemometer and the broader model grid cell, environmental effects such as land use, obstructions, or elevation effects, or the temporal sampling methods of the observational data and the inherent uncertainties associated with that.

Reviewer 1 additionally inquired about whether the observations were corrected for mast flow distortions. Much of the hub-height observational data we worked with did not have the orientation of the anemometers with respect to surrounding structures/towers. Therefore, we did not want to make corrective assumptions and potentially incite additional biases into the observational data. However, we do agree that it is important to acknowledge the limitations of the observational data itself.

**Some detailed comments**

Page 2, line 63:    It should be mentioned here that ERA5 is initial/boundary model in addition to the info in table 1, on page 5.

Done.

Page 10, line 223: Explain "internal variability" and "structure uncertainty" in more detail, and why 10 and 6 ensemble members, respectively, was decided upon.

Thanks for the suggestion. We have added a section particularly for Model uncertainty (section 2.2) to better explain why and how we conduct these uncertainty simulations. The uncertainty quantification section (section 2.4) focus on talking about the bootstrapping and how to express the uncertainty.

Page 14-15, fig. 3:    A legend is missing for the plots.

Added.

Page 15, line 340:  Interpolation in wind direction simply requires conversion of wind direction to components which may be interpolated similar to the wind speeds, and then converted back to wind directions.

Added wind roses for ERA5 data as well.

Page 25, line 489:    Friction velocity is denoted using $u_*$ and not u*.

Fixed.

Page 26, line 503:    Explain why "high friction velocities correspond to weaker winds"

While friction velocity is not a direct scale for wind speed itself, they tend to have strong correlations. Friction velocity quantifies the turbulent momentum flux at the surface. Therefore, higher $u_*$ values correspond to more of the momentum being lost to the surface, leading to weaker wind speeds closer to the ground, especially in areas with high surface roughness. We've added such clarification before discussing the results.

---

## Referee Report (RR1)

**Overall feedback**

The authors have addressed my previous set of comments with care and effort and the manuscript is now in very good condition. Only two minor comments remain that I would like to have addressed.

**Specific comments**

1. Pertaining to the last comment that I provided in round 1: when clicking on the data links in the manuscript PDF, the intended internet pages still do not open properly. I figured out why: the link is between round brackets and the last one is interpreted as part of the URL when I click on it in my online adobe pdf viewer. I think adding a space between the url and the round bracket will solve this.

2. In section 2.5 you have an entire section explaining how uncertainty is quantified and also inter-annual variability. You briefly re-explain this in the captions of Fig. 10, Fig. 11 and Fig. 3. However, I find these explanations in the captions too short and unclear and these actually confused me. I would recommend to remove these and instead refer in the captions to look in section 2.5 on how this is computed. I would then also change the title of section 2.5 to something like: quantification of model uncertainty and inter-annual variability.

---

## Author Response (AR2)

**Overall feedback**

The authors have addressed my previous set of comments with care and effort, and the manuscript is now in very good condition. Only two minor comments remain that I would like to have addressed.

**Specific comments**

1. Pertaining to the last comment that I provided in round 1: when clicking on the data links in the manuscript PDF, the intended internet pages still do not open properly. I figured out why: the link is between round brackets, and the last one is interpreted as part of the URL when I click on it in my online adobe pdf viewer. I think adding a space between the url and the round bracket will solve this.

We have added a space between the first bracket and the beginning of the URL and a space between the last bracket and the end of the URL. Thank you for pointing this out.

2. In section 2.5 you have an entire section explaining how uncertainty is quantified and also inter-annual variability. You briefly re-explain this in the captions of Fig. 10, Fig. 11 and Fig. 3. However, I find these explanations in the captions too short and unclear and these actually confused me. I would recommend to remove these and instead refer in the captions to look in section 2.5 on how this is computed. I would then also change the title of section 2.5 to something like: quantification of model uncertainty and inter-annual variability.

We see how these condensed explanations can be confusing. We have removed the short explanations from the captions for Fig. 10, Fig. 11, and Fig. S3 and instead refer to section 2.5, now named 'Quantification of Model Uncertainty and Interannual Variability', for a more detailed description of the methodology for calculating these metrics.